# Partition complex structure can arise from sliding and bridging of ParB dimers

Lara Connolley [1], Lucas Schnabel[2], Martin Thanbichler [1,2] & Seán M. Murray [1] ✉

In many bacteria, chromosome segregation requires the association of ParB to the *parS*-containing centromeric region to form the partition complex. However, the structure and formation of this complex have been unclear. Recently, studies have revealed that CTP binding enables ParB dimers to slide along DNA and condense the centromeric region through the formation of DNA bridges. Using semi-flexible polymer simulations, we demonstrate that these properties can explain partition complex formation. Transient ParB bridges organize DNA into globular states or hairpins and helical structures, depending on bridge lifetime, while separate simulations show that ParB sliding reproduces the multi-peaked binding profile observed in *Caulobacter crescentus*. Combining sliding and bridging into a unified model, we find that short-lived ParB bridges do not impede sliding and can reproduce both the binding profile and condensation of the nucleoprotein complex. Overall, our model elucidates the mechanism of partition complex formation and predicts its fine structure.

Faithful chromosome segregation is essential for the efficient replication of cells. For this, bacterial chromosomes and low-copy plasmids employ active partitioning systems, with the ParAB*S* system being one of the most widespread[1–3]. This system consists of three components: centromeric-like *parS* sequences and two proteins, ParB which forms dimers that bind specifically to the *parS* sequence, and ParA, an ATPase, the activity of which is stimulated by ParB[4,5].

ParB spreads to several kilobases of DNA surrounding the *parS* sites, which in bacteria are concentrated close to the origin of replication[6]. This spreading is essential in order for these systems to function, though the degree of spreading varies substantially between systems[7–13]. In any case, the result is believed to be a nucleoprotein complex, the partition complex, that is clearly visible using fluorescence microscopy. Originally, spreading was proposed to be due to the formation of a nucleoprotein filament extending out from the *parS* site[7,8,14]. However, it was subsequently shown that there are too few ParB proteins to form such large structures[10]. Instead, ParB was found in vitro to condense DNA through non-specific DNA binding and the formation of protein bridges[10,15–19].

These results motivated modelling studies of partition complex formation. In particular, the spreading and bridging model proposed that the partition complex forms through a combination of long-range (3D) bridging and short-range (1D) nearest-neighbour interactions[20]. However, this model was subsequently argued to be incompatible with the binding profile of ParB from F plasmid[11]. Instead, it was proposed that the observed profile is due to the spatial caging of ParB around the nucleating *parS* site, due to non-specific and transient interactions, and the polymeric nature of the DNA[11,21,22]. This model can also be understood as the weak-spreading limit of the spreading and bridging model[23].

Recently, it has been demonstrated that ParB exhibits *parS*-dependent CTPase activity that is required for correct partition complex formation and dynamics[13,24–28]. CTP-bound ParB dimers were shown to load onto and encompass the DNA at *parS* sites and subsequently slide along the DNA strand before eventually dissociating. It was also shown in vitro that CTP binding allows ParB bridging to occur at physiological concentrations (much lower than that required in the absence of CTP[10,15]) and leads to efficient DNA condensation[29,30]. These results fundamentally change our understanding of how the partition complex is formed and suggest that the previous models need to be reevaluated. In particular, no modelling study has yet provided a unified framework for ParB dimer sliding and bridging. ParB sliding may

[1]Max Planck Institute for Terrestrial Microbiology and Center for Synthetic Microbiology, 35043 Marburg, Germany. [2]Department of Biology, University of Marburg, 35043 Marburg, Germany. ✉e-mail: sean.murray@synmikro.mpi-marburg.mpg.de

also have additional relevance for chromosomal ParABS systems, which typically have several genomically separated *parS* sites and, as a result, more than one peak in the ParB binding profile[9,10,12,13,21,31–34], yet have a single visible partition complex per origin in wild-type cells[10,35].

Here, we investigate the role of ParB sliding and bridging in partition complex formation using semi-flexible polymer and reaction-diffusion simulations. We first show that different ParB bridge lifetimes lead to distinctly different polymer conformations. We then study the short-lifetime regime in which distinct DNA structures (hairpins and helices) form and show how these structures result in the condensation of ParB-coated DNA. We then use stochastic simulations to show that ParB sliding can reproduce the multi-peaked ParB distribution seen experimentally and explore the effects of roadblocks on sliding. Finally, we combine ParB bridging and sliding in out-of-equilibrium coupled polymer/reaction-diffusion simulations and show that bridging does not inhibit ParB sliding for sufficiently short bridge lifetimes. Overall, our work supports a new model of partition complex formation in which ParB dimers load onto the DNA at *parS* sites before sliding diffusively along the DNA. Genomically distant, but spatially proximal, ParB dimers interact to form transient bridges that organise the DNA through the formation of hairpin and helical structures. We speculate that these structures facilitate the additional function of ParB to load Structural Maintenance of Chromosome (SMC) complexes onto the chromosome.

## Results

### Semi-flexible polymer model of ParB bridging

In order to obtain a realistic model of partition complex formation, we use a semi-flexible lattice polymer model of the 10 kb centromeric region of *C. crescentus* in which every monomer corresponds to 20 bp, the approximate footprint of a ParB dimer[28,36] (Fig. 1a). The DNA is modelled as a linear chain using a kinetic implementation of the BFM (Bond Fluctuation Method) polymer model[37]. Since the DNA is stiff at this scale, we introduce an energetic cost for bending to obtain the experimentally measured persistence length of $l_p \sim 120$ bp (Supplementary Fig. 1a)[38]. The BFM is well suited for this as it allows a large set of bond angles[37] and can therefore implement stiffness more realistically than models that only allow 0° or 90° bond angles. The lattice spacing corresponds to 2.2 nm. Allowable monomer moves are attempted at a rate $p$, which defines the fundamental polymeric timescale $\tau = 1/p$ and are accepted according to the corresponding energy cost.

The polymer is constrained by ParB-induced bridging between monomers. This is implemented by allowing any two spatially proximal, non-neighbouring monomers of the polymer to form a bridge at a rate dependent on their ParB occupancies $B_i$, $B_j$ and a rate parameter $k_b$. Each dimer/monomer can only bridge one other dimer/monomer at a time (in the following 'dimer' will always refer to a ParB dimer and 'monomer' to a monomer of the simulated DNA polymer). Bridges dissociate randomly with rate $k_{ub}$ and therefore have exponentially distributed lifetimes. These rates are related to the activation energies for bridging and unbridging, the difference of which, $\Delta E_{ij}$ is the associated binding energy of the interaction (Fig. 1b). Further details of the model are found in the Methods section.

Since ParB dimers can slide along the DNA, the spreading of ParB throughout the centromeric region can occur, at least in principle, independently of any 3D structure. We therefore initially model ParB dimers implicitly, using the relative probability of ParB occupancy, obtained from the experimental binding (ChIP-Seq) profile[12], to specify the probability of a bridge forming when two given monomers come into proximity. This will allow us to first investigate how the observed ParB genomic distribution can, through bridging, affect the structure of the centromeric region, separately from the question of ParB spreading. We will examine the coupling between the two processes of sliding and bridging later.

## ParB bridge lifetime results in distinctly different polymer conformations

The multi-peaked ParB binding profile of *C. crescentus* consists of three clear peaks centred on five *parS* sites (note that two of these *parS* sites are only separated by 42 bp and so are not typically distinguishable in our figures) (Fig. 1c). While two other putative *parS* sites have been identified[12], they are not associated with significant enrichment of ParB. This profile is used in our polymer model to specify the ParB dimer occupancies $B_i$ along the polymer and thus, up to an overall parameter $k_b$, the bridging rate between proximal monomer pairs. Simulating the system, we found a surprising result: ParB-induced bridging leads to two distinct phases for the partition complex. Long bridge lifetimes cause the polymer to collapse into a globule-like structure (Fig. 1f), whereas at shorter bridge lifetimes the polymer is more structured with long extended localised regions of bridging (Fig. 1h). Note that 'long' and 'short' are relative to the polymeric timescale $\tau$. Since we do not have an experimental estimate of this timescale at the lengthscale (20 bp) considered here, we cannot provide specific values.

The effect was also apparent in maps showing the location of the ParB bridges (Supplementary Fig. 1b, d). Whereas bridge maps of the structured conformations show distinctive ±45° lines, those of the globular regime display a more random distribution. However, despite the clear differences in their conformations, both regimes exhibit very similar bridge maps at the population level (Supplementary Fig. 1c, e), which display a checker-board pattern centred on the *parS* sites and have no ± 45° lines detectable. Such a pattern is consistent with an overall preference for contacts within and between regions associated with peaks in the ParB binding profile. A similar pattern was also observed in contact probability maps (Fig. 1g, i), though the globular regime had more contacts for the same number of bridges, as expected from its greater level of compaction. This highlights how the population-average view of DNA organisation may not be informative of the structure of individual conformations.

To better characterise these different regimes we constructed the phase diagram of the system (Fig. 1d). Three regions could be identified: a free coil-like regime in which there are very few bridges (less than 20, a value chosen by inspection) and the polymer behaviour is dictated simply by its stiffness and volume-exclusion (Supplementary Fig. 1f, g), and the structured and globular (unstructured) regimes. We defined the transition between the structured and globular regimes using the ParB weighted radius, i.e. the radius of the *spatial* ParB distribution due to the polymer conformation (see Methods). The globular state has a much smaller ParB weighted radius compared to the structured state with the same number of ParB bridges (Fig. 1e). This radius plateaus as the system goes further into the globular regime. We, therefore, chose a threshold of 55 nm to distinguish the two regimes based on two standard deviations above the plateaued mean value (Fig. 1e).

We propose that these two regimes arise due to the degree of movement that the polymer makes between bridging events. Bridging can be either kinetically limited (limited by the intrinsic bridging/unbridging rates) or diffusion-limited (monomers coming into proximity is the limiting factor and the kinetics are so fast that bridge breaking and forming becomes correlated because newly broken bridges tend to recombine before the polymer can explore the conformational space)[39]. Consider a bridged polymer conformation (Supplementary Fig. 1h). In the diffusion-limited region, recombination effectively increases the bridge lifetime (the activation energy of unbridging)[40,41]. This strengthens a cooperative effect in which new bridges are more likely to form close to an existing bridge because adjacent monomers have a higher likelihood to also be in, or come into, proximity and the time needed for a new bridge to form is much less than than the (lengthened) bridge lifetime. Repetition of this process leads to the extended regions of bridges which we observe

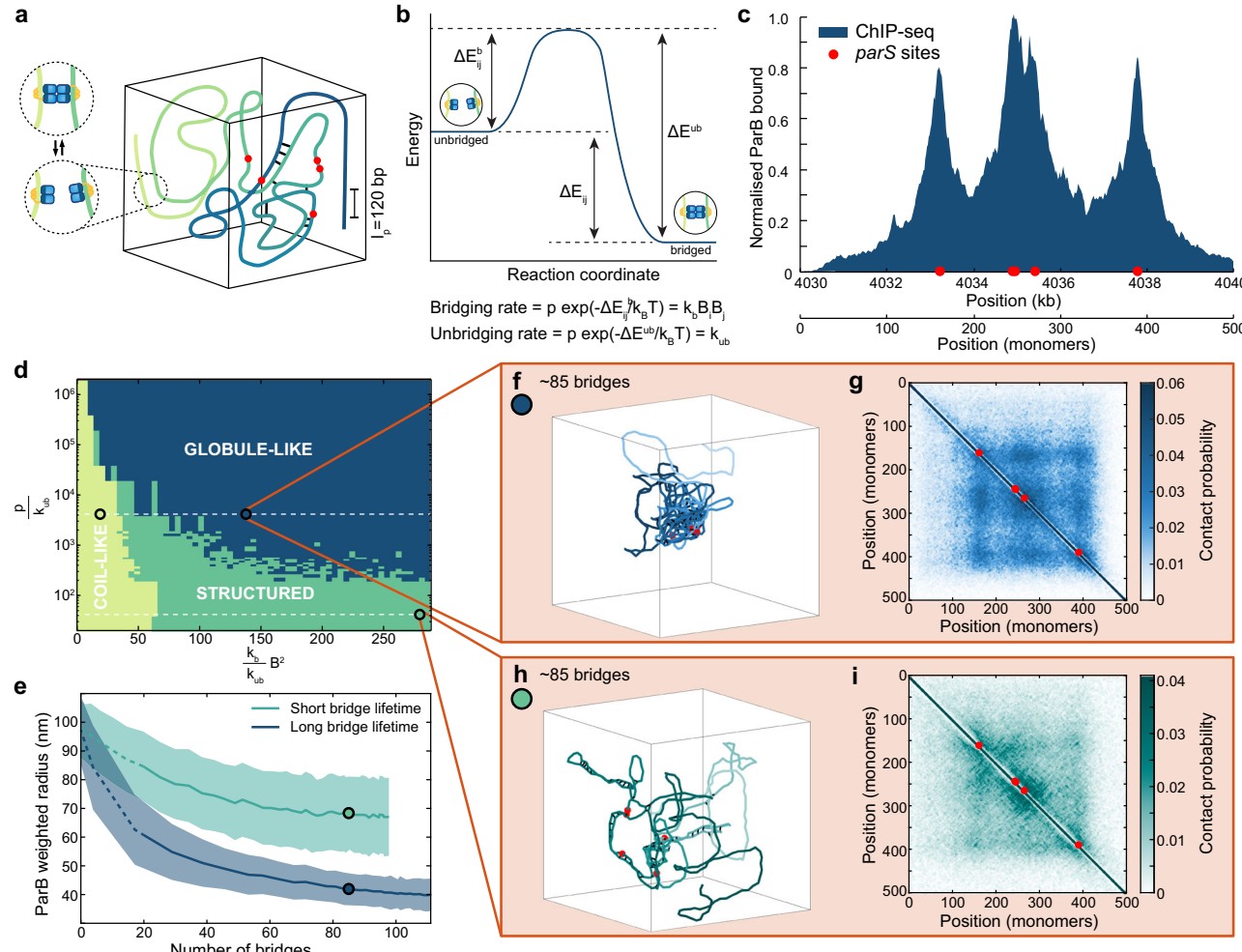

**Fig. 1 | ParB bridge lifetime results in distinctly different polymer conformations.** **a** Schematic of the polymer model. Bridges can form between genomically distant, spatially proximal monomers at a rate proportional to the ParB occupancy at each monomer. The ParB distribution is shown explicitly in **c**. **b** Energy landscape of a bridging–unbridging interaction between two spatially close monomers $i$ and $j$. There is an overall change in energy $\Delta E_{ij}$ upon bridging. The relationship between the activation energies for bridging ($\Delta E_{ij}^b$) and unbridging ($\Delta E^{ub}$) and the reaction rates used in the model are shown. $B_i$ is the ParB occupancy at monomer $i$ and $p$ is the polymeric move attempt rate defining the polymeric timescale $\tau = 1/p$. **c** The normalised ParB ChIP-Seq profile[12] specifies the ParB distribution. *parS* sites are represented by red dots. Note that the two parS sites are very close together. **d** Phase diagram of the system in terms of the effective binding constant $\frac{k_b}{k_{ub}}B^2 = \left\langle \exp(\Delta E_{ij}/k_B T) \right\rangle$, where $B^2$ is defined as the mean $\left\langle B_i B_j \right\rangle$ taken over all $i, j$

with $|i-j| > 1$, and $\frac{p}{k_{ub}} = \exp(\Delta E^{ub}/k_B T)$, the relative bridge lifetime (see Methods). **e** The mean ParB weighted radius for short and long bridge lifetimes (±SD) (indicated by the dashed lines in **d**) as a function of the number of bridges. Data from 1000 conformations for each parameter set. Circles indicate the respective locations of **f**, **g**, and **h**, **i**. **f** An example conformation of the polymer in the globular state. **g** Average contact map at the same location based on 1000 conformations. A contact is defined as two monomers being within five lattice sites of one another. **h** An example conformation of the polymer in the structured state. **i** Average contact map at the same location, otherwise as in **g**. The locations studied in **f**, **g** and **h**, **i** both have an average of ~85 bridges. Equivalent plots for the coil-like regime, indicated by the leftmost circle in **d** are shown in Supplementary Fig. 1f, g. Source data are provided as a Source Data file.

(Supplementary Fig. 1h). The conformational cost of bridging is also reduced by having bridges clustered together. A similar effect has previously been seen in simulations of the bridging protein H-NS[42]. On the other hand, at long bridge lifetimes bridging events are kinetically limited and the polymer is able to reorganise and explore the conformational space between bridging events. As a result, there are many more potential bridging events (monomers coming into proximity) away from existing bridges than in the short-lifetime diffusion-limited regime. This results in both more bridges and a more random distribution of bridges (Supplementary Fig. 1b) and hence a globular polymer configuration (Supplementary Fig. 1h). The increase in the number of bridges overcomes the additional conformational cost of having the bridges dispersed rather than localised as in the structured regime.

Since the globular regime is reminiscent of previous proposals for partition complex organisation[11,20,21], we will focus next on examining

the structured state. We will return to the globular state in the final section.

## Short-lived ParB bridging leads to the formation of hairpins and helices

The structured regime found at short ParB bridge lifetimes is defined by the presence of two distinct structures, hairpins and helices. Hairpins form by the polymer bending 180° back on itself to form bridges between anti-parallel segments, whereas in helices, the polymer turns a full 360° with bridges between parallel segments (Fig. 2a). These two structures are visually different but also have different underlying bridging patterns which allows them to be clearly identified in bridge maps. Hairpins correspond to +45° lines whereas helices correspond to −45° lines. The location of the line relative to the main diagonal indicates the length of the loop of the hairpin or the period of the helix. Unsurprisingly, these structures generally form near the *parS* sites.

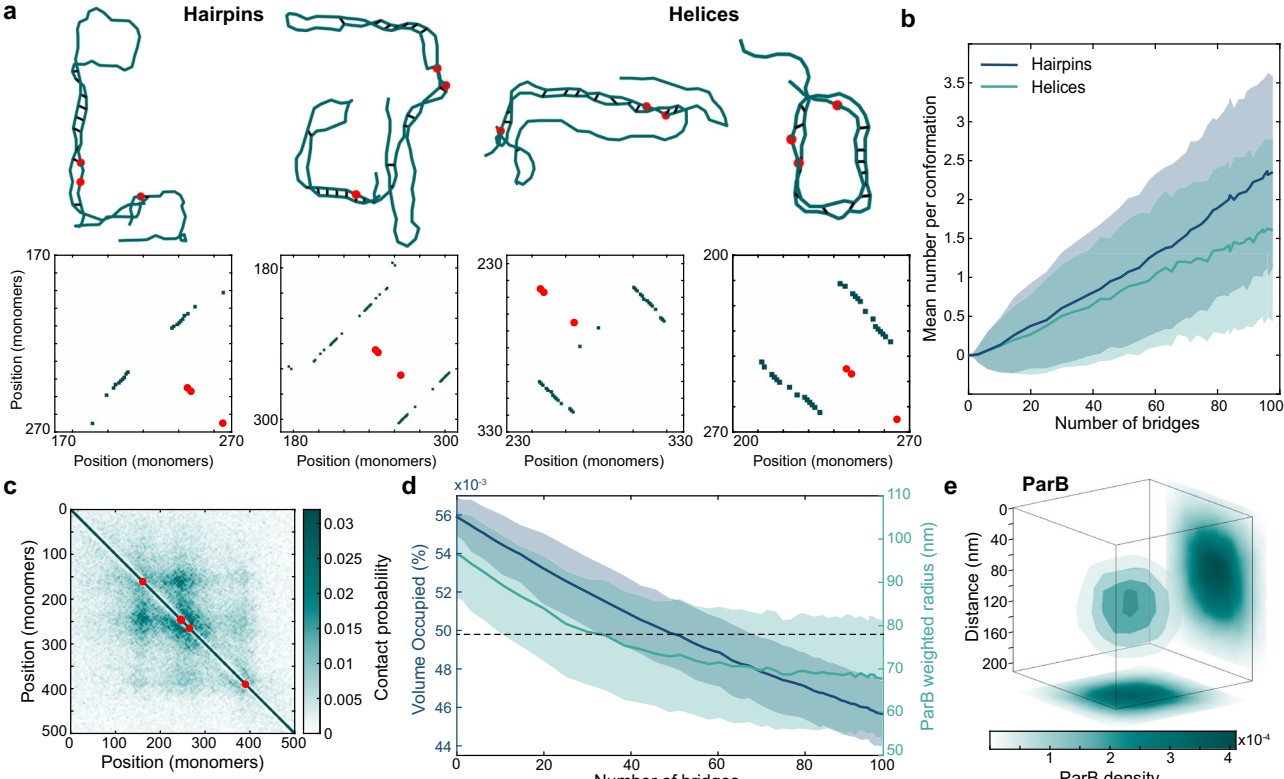

**Fig. 2 | Short-lived ParB bridges results in the formation of hairpins and helices.** **a** Example structures with corresponding bridge maps (bridge maps have been dilated to make lines clearer) for hairpins and helices with an average of 30 bridges. Full polymer conformations and bridge maps can be seen in Supplementary Fig. 2a. Red dots indicate the *parS* sites. Note that as two of the sites are only separated by 42 bp they are not always distinguishable. **b** Mean number of hairpins and helices per conformation. Shading represents the SEM. **c** Average contact map for the polymer at an average of 30 short-lived bridges. **d** Mean volume occupied by the polymer and the mean ParB weighted radius. Shading represents the SD. The dotted line at 78 nm shows the experimentally determined ParB radius for *C. crescentus*. **e** Three-dimensional ParB density from partition complexes with an average of 30 bridges showing a radius of 78 nm. Source data are provided as a Source Data file.

However, we observed substantial variation: the tip of a hairpin (indicated by where the 45° line in the bridge map intersects the main diagonal) was often reasonably far from the nearest *parS* site (Fig. 2a). At lower levels of bridging, these structures most frequently form within the region covered by the central peak containing three *parS* sites.

We made use of the distinctive ±45° lines to quantify the occurrence of hairpins and helices as a function of the degree of bridging in the system (Fig. 2b). We found that the frequency of both structures increased approximately linearly with the number of bridges, with hairpins being the most common. From ~30 bridges every conformation contained at least one structure (Supplementary Fig. 2b). At the highest levels of bridging studied (~100 bridges) each conformation contained 3–4 structures, which could be of either type and involve multiple and distant *parS* sites. Nevertheless, the different constituent structures could still be identified from the ±45° lines in the bridge maps. However, as discussed in the previous section, the ±45° lines are not apparent in the ensemble average contact map or bridge map which displays a checkered pattern centred on the *parS* sites (Fig. 2c and Supplementary Fig. 2c).

Consistent with in vitro experiments, ParB bridging led to the condensation of the DNA polymer. Both the volume occupied (see Methods for definition) and the squared radius of gyration decreased with the number of bridges (Fig. 2d, Supplementary Fig. 2d). In vivo the nucleoprotein complex is visualised through the spatial distribution of a fluorescently tagged variant of ParB, which forms distinct foci within cells. To connect with these observations, we combined the genomic distribution of ParB on the DNA (based on the ChIP-seq profile), with our simulated conformations of the DNA polymer to obtain the resulting spatial distribution of DNA-bound ParB (Fig. 2e). The resultant spherical density was reminiscent of that observed experimentally using single-molecule microscopy. The radius of the partition complex of *C. crescentus* has been measured experimentally using super-resolution PALM microscopy to be ~78 nm[35]. This could be achieved in our simulations with just 30 ParB bridges. This corresponds to a 20% decrease compared to the value in the absence of bridging (Fig. 2d).

## ParB sliding model can reproduce the multi-peaked ChIP-seq profile

In the previous sections, we ignored the question of how the genomic distribution of ParB is formed but rather focused on how the observed distribution can affect, through bridging, the organisation and compaction of the centromeric region. In this section, we do the opposite and consider how ParB spreads along the DNA, while ignoring any potential effect of ParB bridging. Several recent in vitro studies have shown that ParB dimers of chromosomal ParABS systems (*C. crescentus*, *Myxococcus xanthus* and *Bacillus subtilis*) can entrap DNA at *parS* before sliding away in either direction in a manner akin to a DNA clamp[13,24–28,30]. Dissociation is believed to be primarily due to CTP hydrolysis. We recently developed a stochastic model of this spreading mechanism in the context of *M. xanthus*[13] and found that loading at *parS* sites, 1D diffusion along the DNA and dissociation was indeed able to qualitatively reproduce the observed ParB binding profile from ChIP-Seq. The predicted 1D diffusion coefficient also agreed with single-molecule microscopy measurements. However, the binding profile of *M. xanthus* is relatively noisy

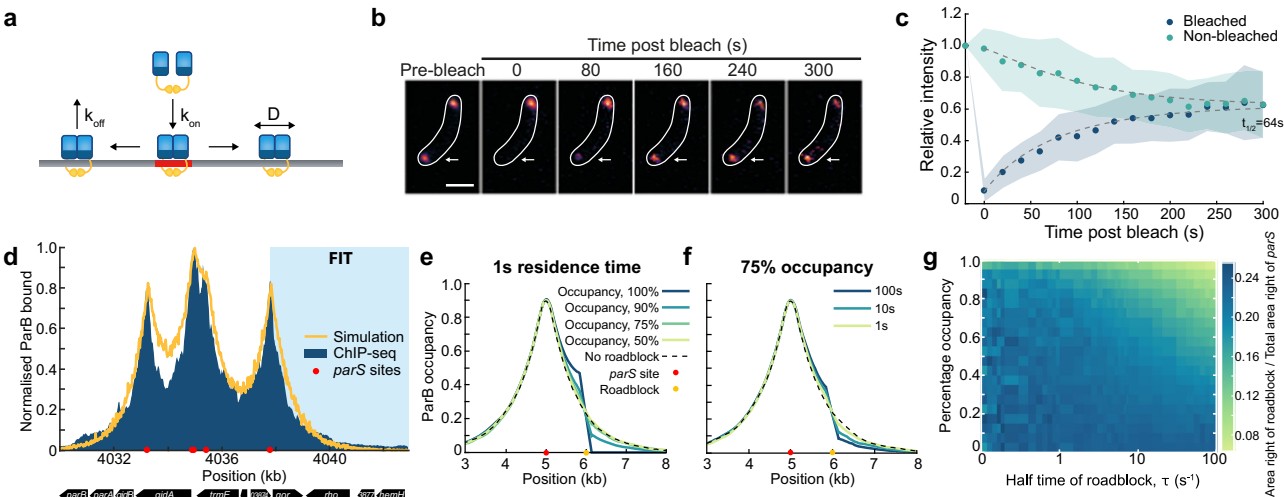

**Fig. 3 | ParB sliding can reproduce the multi-peaked *C. crescentus* profile.**
**a** Diagram representing the sliding model, showing ParB dimers binding at a *parS* site, diffusing along the lattice and unbinding. **b** Representative images of fluorescence recovery after photobleaching (FRAP) experiment. A single eGFP-ParB focus (arrow) was photobleached in a cell containing two foci. Scale bar is 1 μm. All timelapse images for this cell can be seen in Supplementary Fig. 3a. **c** Analysis of FRAP data. The average relative intensity of the bleached (blue) and unbleached (green) partition complex are shown as a function of time ($n = 51$ cells). Dashed lines represent the behaviour from a fitted model, which found a half-time of 64 s (see Methods). Shading represents the SD. **d** Simulation of ParB sliding compared to ChIP-seq data from ref. 12, both normalised by maximum height. Shaded area indicates the part of the ChIP-seq profile that was fitted to an exponential to find the effective diffusion coefficient. **e** Simulations of ParB sliding at 1 s residence time for different roadblock occupancies. The roadblock is indicated by the yellow dot. Similar figs. for 10 and 100 s residence times are in Supplementary Fig. 3d. **f** Same as in **e** but for 75% occupancy with a varying roadblock lifetime. In both **e** and **f** the dotted line shows the profile when there is no roadblock. **g** Phase diagram displaying the difference between simulations with and without a roadblock. The colour indicates the ratio of the area of the distribution to the right of the roadblock to the total area on the right-hand side of the *parS* site. The locations of the roadblock and the *parS* site are shown in **e** or **f**. Source data are provided as a Source Data file.

and consists largely of a single peak centred on a cluster of all but one of its 24 *parS* sites. Therefore, the multi-peaked and less noisy profile of *C. crescentus* may serve as a better test of the in vivo relevance of the loading and sliding model. While other multi-peaked ParB binding profiles are available[21,32–34], the binding affinity of each *parS* site of *C. crescentus* has been determined so that ParB loading in the model can be described using a single parameter rather than one for each *parS* site.

We use the same fundamental model as previously[13], modified for *C. crescentus*. ParB dimers load onto the DNA, modelled as a 1D lattice, at any of five *parS* sites. Loaded dimers then diffuse along the lattice with effective diffusion coefficient $D$ and dissociate randomly at a rate $k_{off}$ (Fig. 3a). Previous experimental studies have shown that upon loading at *parS* sites ParB dimers form a protein clamp that completely encompasses and subsequently slides along the DNA strand[13,24–28]. Consistent with this, it has been shown that ParB dimers are unable to move past DNA-bound roadblocks[25,27]. Note that due to the relatively tight entrapment of the DNA strand, we assume in our model that sliding ParB dimers act as obstacles for one another (based on the structures one dimer is not expected to be able to pass through the loop of another[13,25,26]). The *parS* sites must also be free for a dimer to load. The total number of ParB dimers is fixed at the measured value of 360[35], with unbound dimers treated as a well-mixed bulk (cytosolic) population. The relative loading rate at each site is specified by its relative affinity for ParB[12], leaving a single overall loading rate $k_{on}$.

We first determine the effective diffusion coefficient $D$ and the dissociation rate $k_{off}$. To estimate the latter, we performed fluorescence recovery after photobleaching (FRAP) of eGFP-ParB foci in predivisional cells containing two foci (partition complexes) (Fig. 3b). After bleaching one of the two foci, the fluorescence signal recovered with a half-time of 64 s (Fig. 3c, Supplementary Fig. 3a, b). This provides an estimate for the dissociation rate $k_{off}$ (see Methods). Interestingly, roughly similar ParB recovery times have been measured for *M. xanthus*[13] and F plasmid[21,43].

To determine the diffusion coefficient, we fit the outer part of the third peak to an exponential $e^{x/\lambda}$ with $\lambda = \sqrt{\frac{D}{k_{off}}}$ (Fig. 3d), the predicted continuum distribution under this model for an isolated *parS* site (see Methods). The fitted value of $\lambda = 710$ bp, then gives $D = 5600$ bp² s⁻¹ = $6.1 \times 10^{-4}$ μm² s⁻¹. This is lower than previously reported diffusion coefficients for proteins diffusing along bacterial DNA[44] including previous measurements of ParB[13,43], which are on the order of $10^{-2}$ μm² s⁻¹. We will see below that a larger value is required in the coupled model to overcome the roadblock effect of bridges.

One model parameter remains to be determined - the overall loading rate of ParB $k_{on}$. Previous measurements have estimated that ~80% (290) of ParB dimers in the cell are in ParB foci[35]. In contrast, we find that even at high loading rates <220 ParB dimers are associated with the DNA (Supplementary Fig. 3c). Increasing $k_{on}$ further does not substantially increase the number of ParB bound as the *parS* sites are almost continuously occupied. The disparity in the number of DNA-associated dimers may be due to several factors. Firstly, the maximum possible number of associated dimers in our simulations is dependent on the chosen discretisation since each lattice site/monomer can be bound by a single ParB dimer. Thus if the footprint of ParB is smaller than our discretisation size of 20 bp, we would be underestimating the achievable ParB occupancy. Secondly, the in vivo estimate of the cellular ParB concentration is based on quantitative Western blotting, which has a substantial margin of error[45]. ParB foci may also contain a cytosolic or non-specifically bound population that is not accounted for in our model.

Given the above, we choose the loading rate for our model by finding the best fit of the simulations to the ChIP-seq data (Supplementary Fig. 3c), obtaining $k_{on} = 200 \cdot k_{off}$. This results in remarkably good agreement between the model and the ChIP-Seq profile (Fig. 3d), indicating that loading and diffusive sliding of ParB dimers can indeed explain the observed binding profile. It also suggests that dimers are largely unaffected by transcription and other processes that could hinder ParB spreading since we have not accounted for these effects in

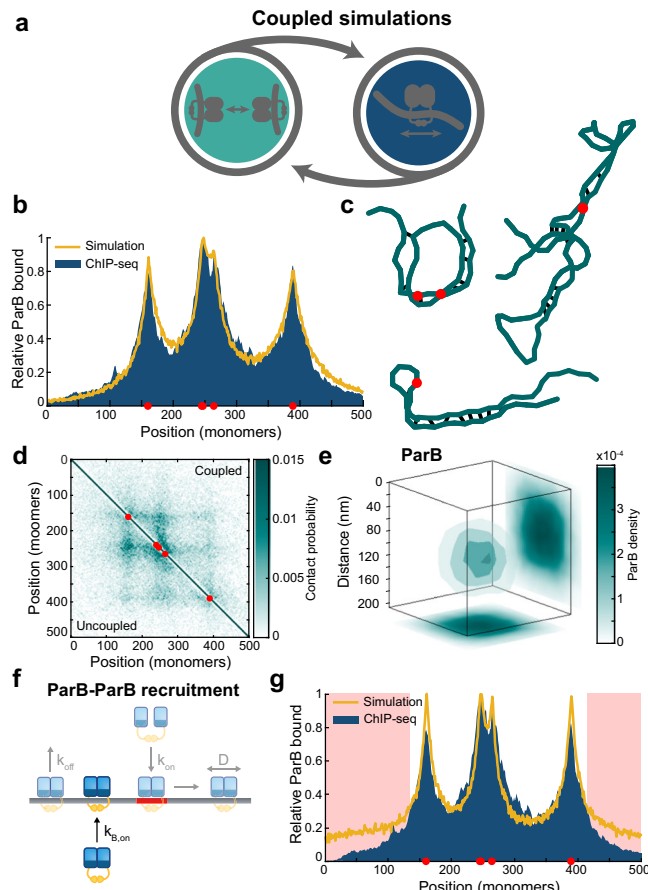

**a** Coupled simulations

**b**

**c**

**d**

**e** ParB

**f** ParB-ParB recruitment

**g**

**Fig. 4 | Sliding is not inhibited by short-lived ParB bridges. a** Representation of coupled polymer simulations in which we combine bridging and sliding. **b** Profile of ParB as generated from sliding and bridging simulations with a bridge lifetime of 1s compared to ChIP-seq data from a previous study[12], both normalised by maximum height. **c** Examples of hairpin and helical structures found in coupled simulations, with an average of 25 bridges. Full conformations and individual bridge maps can be found in Supplementary Fig. 4a. **d** Average contact map for coupled simulations (top right) and uncoupled simulations (bottom left) with a radius of ~78 nm. Each contact map is made from 1000 simulations. **e** Three-dimensional average of the ParB partition complex for 25 bridges with a radius of 78 nm. **f** Diagram displaying the addition of ParB-ParB in cis recruitment to the coupled model. **g** Profile of ParB along the polymer with additional ParB–ParB recruitment, areas where the profile is substantially different to ChIP-seq data are highlighted in red. For all plots shown the simulations are in the structured regime. Source data are provided as a Source Data file.

our model. This may not be the case for other systems such as F plasmid that show changes in the binding profile coincident with promoters[21,46]. Indeed, in vitro experiments have shown that high-affinity DNA-binding proteins, such as EcoRI (with the catalytically-inactive E11Q mutation) and TetR, can block the sliding of ParB dimers along the DNA[25,27,30].

### Residence time and percentage occupancy of roadblocks impacts their effect on ParB sliding

To better understand how roadblocks can impact the spreading of ParB dimers, we used our sliding simulation to examine the effect on spreading from a single *parS* site. Representative of the biological situation, we do not consider a permanent roadblock but rather a dynamic one, which we specify in terms of its average lifetime and occupancy i.e. the overall fraction of time that the roadblock is bound to the DNA. These two measures are independent of each other, for instance, a roadblock that is present and absent for 1s at a time has the

same 50% occupancy as a roadblock present and absent for 10s at a time. We found that at a lifetime of 1s, the roadblock had a surprisingly mild effect on spreading, only becoming noticeable from an occupancy of ~75%. Even at 95% occupancy, roughly half the number of dimers slide past the site of the roadblock as in its absence (Fig. 3e). Similarly, at 75% occupancy, a negative effect on spreading was only observed for roadblock lifetimes < ~1s (Fig. 3f).

We can understand these results as follows. When the roadblock is present for a time much shorter than the time interval between dimer crossing attempts then a backlog of dimers does not develop. Even for longer times, the backlog of dimers can be cleared if there is enough time between roadblock events i.e. if the average roadblock occupancy is sufficiently low (Fig. 3g). These results may explain why we observe no significant deviation of the ParB binding profile from that expected from our simple loading and sliding model—the in vivo occupancy and residence times of proteins binding to the centromeric region of *C. crescentus* may simply not be large enough to substantially affect ParB spreading.

### Coupled simulations of sliding and bridging

We next investigate whether ParB bridging is compatible with the ParB binding profile i.e. would the spontaneous formation of ParB bridges between spatially proximal but genomically distant ParB dimers limit overall ParB spreading and produce a fundamentally different binding profile? To answer this question we coupled our polymer and sliding simulations together (Fig. 4a). Unlike the previous simulations, the bridging of proximal monomers is now explicitly dependent on the presence of a ParB dimer at each site rather than on a pre-specified ParB distribution. Note that since ParB sliding is a non-equilibrium process, this coupled model is therefore necessarily out-of-equilibrium. We assume that bridged dimers are not able to slide along the DNA, due to the entrapment of genomically distant regions, so they act as roadblocks for unbridged sliding dimers. The simulations are run until steady state and the ParB distributions and polymer conformations recorded.

The same values determined in the previous section are used for ParB dimer loading ($k_{on} = 200 \cdot k_{off}$) and dissociation ($k_{off} = \frac{\log(2)}{64}$ s$^{-1}$). There are currently no estimates for the bridge lifetime. We expect bridges to have a significantly shorter lifetime than that of ParB dimers on the DNA and therefore a nominal value of $k_{ub} = 1$ s$^{-1}$ is chosen. With too high a value (of the order of the ParB lifetime on the DNA) sliding ParB dimers would not have time to move past roadblocks (ParB bridges) before unbinding. We access the two regimes discussed in the first section through the mobility of the polymer. We arbitrarily choose two values of *p* to represent the globule-like and structured regimes (based on the sweep of the simple bridging model). This leaves the sliding diffusion coefficient and overall bridging rate as free parameters. These are chosen such that we can reproduce both a 78 nm ParB radius and the expected genomic distribution. We are unable to use the value found for the diffusion coefficient in the previous section due to the introduction of ParB bridges resulting in ParB dimers sliding a shorter distance creating sharper peaks. Thus this value must be tuned based on the number of ParB-ParB bridges. For the structured regime discussed below a value of $4.4 \times 10^{-3}$ µm$^2$ s$^{-1}$ is used to resolve the ChIP-seq profile for 25 bridges. This is a lower bound. Larger values have very little effect on the ParB profile recovered as sliding ParB dimers reach equilibrium between bridging events. This lower bound is within an order of magnitude of the diffusion coefficient of ParB measured in vivo[13,43].

For the structured regime, we found that the coupled simulations could reproduce the binding profile measured by ChIP-seq (Fig. 4b), with an even better fit than we obtained from the non-polymeric sliding simulation (Fig. 3d). Importantly, we also observed the same hairpin and helical structures as in the uncoupled polymer simulations that had the ParB binding profile given as a input (Fig. 4c) and obtained

very similar average contact (Fig. 4d) and bridge maps (Supplementary Fig. 4b). These structures again compact the polymer and we could achieve the measured radius of 78 nm (Fig. 4e).

In the globular regime we were able to broadly reproduce the ChIP-seq profile although the simulations could not accurately capture the depth of the valleys (Supplementary Fig. 4c). Similar contact and bridge maps were found (Supplementary Fig. 4d, e). Interestingly, the partition complex is less condensed in the coupled simulations than in the uncoupled simulations at the same mean number of bridges, whereas no significant difference was detected for the structured regime (Supplementary Fig. 4f, g).

Recent in vitro studies have shown that DNA-loaded ParB dimers of *B. subtilis* can load additional dimers independently of *parS* ('ParB-ParB recruitment')[47], potentially explaining the cooperative non-specific DNA binding observed previously[15,18,19] and consistent with interactions between dimers through their N-terminal domains[16,48]. To explore whether such recruitment could be relevant in vivo, we added *in cis* ParB-ParB recruitment to our model (Fig. 4f). Although in trans recruitment was also shown by the same authors this would be substantially more challenging and computationally intensive to implement.

We found that even a relatively low ParB-ParB recruitment rate, for which the total number of bound dimers increased by less than 20%, results in a fundamentally different binding profile. ParB spreading was increased through the appearance of slowly decaying 'shoulders' at the extremes of the distribution. As a result the distinctive exponential decay seen in the experimental ChIP-seq profile is no longer reproduced (Fig. 4g) and this could not be remedied by changing the model parameters. This result suggests that ParB-ParB recruitment does not play a significant role in vivo in ParB spreading, consistent with the finding of Tišma et al. that ParB-ParB recruitment accounts for only 10% of ParB loading events in vitro[47].

## Discussion

The sliding and bridging model presented here uses recent discoveries to probe the formation and structure of the partition complex. Recent in vitro-based studies have shown that, dependent on CTP, ParB can load onto DNA at *parS* sites before sliding randomly along the strand[13,24–28]. It was also shown that ParB can efficiently condense DNA, again in the presence of CTP, through the formation of bridges between genomically distant DNA regions[10,15,29,30]. While we have not explicitly modelled the CTP-dependent nature of these processes, our model is consistent with CTP hydrolysis triggering the unbinding of ParB dimers and therefore setting the length scale of sliding[13,25]. Our model predicts that the dynamic sliding and bridging of ParB results in two different conformational regimes, one globular, one structured for long and short bridge lifetimes respectively. The latter regime is dependent on the stiffness of the DNA. If that is ignored, short-range bridges between next to neighbouring monomers dominate and DNA structures do not develop. This is consistent with the results of previous studies of chromatin organisation[49,50] that do not incorporate stiffness and with bridging by much larger molecules[42]. We also showed how the genomic distribution of ParB could define its spatial distribution through the formation of ParB bridges. We then explicitly modelled both the sliding of ParB along the DNA and the formation of ParB-ParB DNA bridges. Importantly, we found that sufficiently short-lived bridges do not hinder sliding of ParB along the DNA and our model could reproduce both the measured genomic and spatial distribution of *C. crescentus* ParB.

We speculate that the two different regimes could have relevance in different biological contexts. The hairpins and helices of the structured regime may facilitate the loading of SMC (structural maintenance of chromosomes) complexes onto the DNA[51]. While this is known to be due to ParB at the *parS* sites[52], the precise mechanism is a topic of ongoing study[53,54]. However, ParB mutations that eliminate

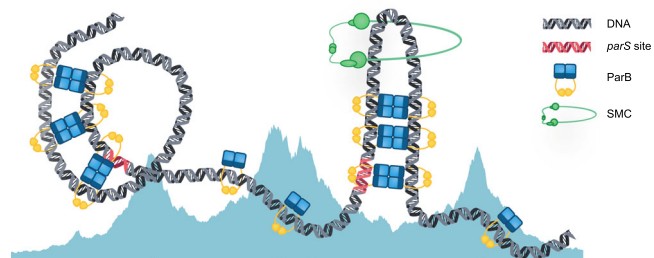

**Fig. 5 | A sliding and bridging model can reproduce the genomic and spatial distribution of ParB, forming hairpins and helical structures which organise the DNA.** Cartoon of the structure of the partition complex. ParB dimers load at the *parS* sites and then slide along the DNA where they can interact with genomically distant, spatially proximal dimers to form ParB-ParB bridges. These bridges can organise the DNA into hairpin and helical structures. Hairpin structures could potentially be involved in the recruitment of SMC onto the DNA.

SMC recruitment are also known to reduce the ability of ParB to form a higher-order nucleoprotein complex[53,55,56]. This leads us to postulate that the ParB-induced DNA structures we observe are relevant for the loading of SMC complexes (Fig. 5). Furthermore, chromosomal ParABS systems often have multiple separated *parS* sites[6] that produce a multi-peaked binding profile[9,10,12,13,21,31–34], whereas a single cluster of *parS* sites appears to be more common for plasmid-based systems[57]. Separated *parS* sites would allow the formation of multiple hairpins and could thereby be beneficial for SMC loading.

In contrast, ParABS-carrying plasmids, especially those of E. coli and other bacteria that do not carry SMC[58], would likely not require these structures. Instead it may be advantageous to form a more compact partition complex to better facilitate the partitioning function of ParABS. Indeed, while F plasmid ParB spreads over a four times larger region than ParB of *C. crescentus*[11], the resultant partition complex is significantly smaller (a radius ($2\sigma$) of 35 nm)[43]. Thus, we speculate that plasmid-based ParABS systems may operate in the more compact globular region.

The bacterial chromosome is on average negatively supercoiled[59]. This results in the formation of supercoiled loops (plectonemes) that partition the chromosome into topologically isolated ~10 kb domains[60]. However, in vitro DNA experiments and simulations have also detected simple plectonemes in the low kilobase range[61–63]. Thus, these structures, which are topological similar to the hairpins, may be relevant at the 1–2 kb scale of the peaks of the ParB distribution. We expect they would only promote bridge formation due to bringing DNA strands into contact (helices are likely less relevant as they are thermodynamically disfavoured compared to plectonemes[64]). Indeed, a previous model of F plasmid partition complex formation argued that supercoiling is required to explain the observed compactness of the partition complex of that system[22].

The conformations we observe in our simulations are similar to those recently seen using atomic force microscopy for *B. subtilis* ParB[30] but more detailed study is required to test our prediction of hairpin structures. Our model could also be better characterised by knowledge of the ParB-ParB bridge lifetime, which could be achieved in vitro by using magnetic tweezers to probe the relaxation time of ParB-condensed DNA upon removal of ParB from the buffer. In vivo characterisation of the partition complex is more challenging. While our simulated contact maps are in principle comparable to the experimental contact maps produced by chromatin conformation capture (HiC), the resolution of this technique is not yet sufficient to probe DNA structure at the short lengthscale of the *C. crescentus* centromeric region. This may change as the technique improves[65,66].

Overall, we have presented a physical model for the formation of the partition complex of the ParABS system. Our dynamical sliding and bridging model reconciles the recent result that ParB spreads

along the DNA by sliding like a DNA clamp with the ability of ParB to condense the centromeric region into a nucleoprotein complex. Future experimental work will help in evaluating the model and testing its predictions.

## Methods

### Polymer model

We simulate the DNA of the centromeric region using the bond fluctuation model (BFM)[37], a lattice polymer model that reproduces Rouse polymer dynamics. Specifically, the DNA is modelled as a linear chain on a 3D cubic lattice with reflective boundary conditions. Each monomer occupies a cubic site of the lattice including the eight associated lattice points. Furthermore, each lattice point can only be occupied by one monomer at a time to account for the excluded volume of the chain. Individual monomers are connected by bond vectors taken from a set of 108 allowed vectors. This set is chosen such that the polymer chain cannot pass through itself[37]. Monomers can move one lattice site at a time in each Cartesian direction subject to the constraint on allowed bond vectors and the excluded volume. The model is ergodic in that the configuration space of the polymer can be sampled using only local moves. We use a kinetic implementation, based on the Gillespie method[67], as this allows us to incorporate the dynamics of bridging (see below).

We take each monomer to represent 20 bp since this is the approximate footprint of a ParB dimer and leads to computationally tractable simulations. In *C. crescentus*, ParB spreads over a ~10 kb region of the chromosome and we therefore simulate a polymer with a corresponding length of 500 monomers.

In order to account for the stiffness of the DNA, we follow the approach of Zhang et al.[68] and introduce a squared-cosine bending potential $E$ between successive bonds

$$\frac{E}{k_BT} = k_s(1 - \cos\theta)^2, \qquad (1)$$

where $\theta$ is the change in angle between successive bonds and $k_s$ is a parameter controlling the stiffness. Note that a monomer move affects three bond angles: the angle at the monomer and at its two neighbours. Attempted moves are then accepted with probability $P = \min(1, \exp(-\Delta E/k_BT))$, where $\Delta E/k_BT$ is the change in energy due to the move. Allowable monomer moves (moves that obey the volume exclusion, bond length, bridge length conditions) are attempted at a rate $p$. We set the stiffness parameter $k = 14$ to give a persistence length (calculated according to the angle between consecutive bonds[68]) of 120 bp (Supplementary Fig. 1a) in line with experimental measurements[38].

Similar to other bacteria, chromosomal DNA in *C. crescentus* constitutes a volume fraction of ~1–2%. We obtain this volume ratio in the simulations by setting the size of the lattice appropriately. In the BFM the volume occupied by the polymer is not a fixed quantity due to the large set of bond vectors—the excluded volume associated with each monomer can overlap. However, we can measure the occupied volume by dilating the three-dimensional binary image describing the occupancy of each lattice site using a cubic structuring element of width 3 (we use the MATLAB function *strel*). This gives precisely the excluded volume of the entire polymer (recall that each monomer uniquely occupies eight lattice points). Using a $90 \times 90 \times 90$ cubic lattice and the stiffness parameter chosen above, we find an excluded volume fraction of 1.7% (an excluded volume per monomer of ~22 lattice sites).

Upon this stiff polymer framework, we implement bridging between non-neighbouring monomers. Our implementation is similar to that of Bohn and Heermann[69]. Any two non-neighbouring monomers that are within (strictly less than) a spatial distance of 3 lattice sites are allowed to bridge. The rate (probability) of bridging depends on the positions of the monomers within the polymer. In Figs. 1 and 2, the rate is specified, up to an overall factor, by the ParB binding profile (determined by binning the experimental ChIP-Seq profile at the 20 bp resolution of the polymer). The rate of bridge formation between two monomers that are in proximity is then equal to the overall bridging rate, $k_b$ multiplied by the product of the ParB occupancy at each site, $B_iB_j$. Bridged monomers can still move on the lattice but must maintain a bridge length strictly less than 3. Each monomer can only bridge with at most one other monomer.

Bridges break randomly with a mean lifetime $1/k_{ub}$ and a value of 1 s is used throughout this manuscript. The timescale of monomer dynamics $\tau = 1/p$ is not experimentally known at the short lengthscales simulated here. In the first part of this paper, only the ratio of these two timescales is relevant such that we can leave $k_{ub}$ fixed and vary the move rate $p$ of the polymer. For the phase diagram presented in Fig. 1d we vary $p$ from 20 to $2 \times 10^6$, for the rest of the paper we take the arbitrarily chosen values of $p = 40$ to represent the structured regime and $p = 4 \times 10^3$ for the globular regime, shown by the white dotted lines in Fig. 1d.

For any given parameter set, simulations are first run until equilibrium is reached as determined by the volume occupied by the polymer reaching an approximate constant value. We use the volume occupied rather than the usual squared radius of gyration as the former was found to be a much less noisy measure. The conformation of the polymer is then recorded. We repeat this process for 1000 random initial configurations.

**Calculating ParB radius.** The ParB radius is calculated by combining the genomic distribution of ParB on the DNA (either based on the ChIP-seq profile for the uncoupled polymer simulations or the simulated position of ParB dimers in the coupled simulations) with the simulated conformations of the DNA polymer to obtain a spatial distribution of DNA-bound ParB. We take an average across all 1000 conformations, aligning them by their centroids, to obtain a 3D density. We then determine the radius within which 95% of ParB dimers are found. We convert this value from lattice units to nanometres as follows. In our (stiff) polymer simulations, the bond length between monomers varies but has an average value of 3.0 lattice units. Since every bond/monomer corresponds to 20 bp and the length of a base pair is 0.33 nm[70], a lattice unit corresponds to 2.2 nm.

### Model of ParB sliding

We model the loading, sliding (diffusion) on and unbinding of ParB dimers from the DNA using the same approach as in our recent work on *M. xanthus*[13]. The DNA is modelled as a one-dimensional lattice with each lattice site corresponding to 20 bp. The model is single occupancy—loading and sliding can only occur if the target lattice site is free. ParB dimers can load at some number of special lattice sites, corresponding to the *parS* sites. For the simulations of sliding in *C. crescentus*, the relative loading rate at each *parS* site is determined by $\frac{1}{K_d}$ where $K_d$ is the measured dissociation constant[12]. The loading rate at each site is then determined by multiplying by an overall factor $k_{on}$. Dimers diffuse to unoccupied neighbouring lattice site with a rate $d = D/h^2$, where $D$ is the effective diffusion coefficient and $h$ is the lattice spacing. Unbinding occurs randomly with rate $k_{off}$. The total number of dimers is fixed as 360 as estimated for *C. crescentus*[35]. Any unbound dimers are assumed to be in the cytoplasm which we take to be well-mixed. This is a simplifying assumption and we make no claims regarding the mechanism of ParB targeting to the *parS* sites. Evidently, it must be fast enough relative to the measured 64 s turnover of the partition complex. This might be facilitated by the high local concentration of bound dimers near the *parS* sites (in which case the cytosol would not actually be well-mixed). From the point of view of the model, this would only mean a lower loading rate $k_{on}$ would suffice to load the same amount of ParB.

For each parameter set the simulation is first run until steady state is reached and then the distribution of ParB is recorded at regular time intervals, sufficiently separated to be independent samples of the steady-state distribution.

**Analytical description.** We provide an analytical description of sliding for the simplified case of a single parS site. Consider ParB dimers diffusing on an infinite single-occupancy lattice (lattice spacing $h$). Dimers can move to any unoccupied neighbouring site at a rate $d$. Dimers load onto the lattice at a site $i = 0$ with rate $\bar{k}_{on}$ and unbind with rate $k_{off}$. We denote the probability of there being $n$ ParB dimers at the $i$th site by $P_n(i, t)$ ($n = 0, 1$ due to single occupancy). The chemical master equation which corresponds to this system of reactions is

$$\frac{\partial P_1(i,t)}{\partial t} = dP_1(i-1,t)P_0(i,t) + dP_1(i+1,t)P_0(i,t) - dP_0(i-1,t)P_1(i,t)$$
$$- dP_0(i+1,t)P_1(i,t) - k_{off}P_1(i,t) + \bar{k}_{on}\delta_{i0}P_0(i,t). \tag{2}$$

Using $P_0(i, t) + P_1(i, t) = 1$, we can rewrite this in terms of the expected number of dimers at each site, $E_i(t) = \sum_{n=0}^{1} nP_n(i,t) = P_1(i,t)$, as

$$\frac{\partial E_i(t)}{\partial t} = d(E_{i-1}(t) + E_{i+1}(t) - 2E_i(t)) - k_{off}E_i(t) + \bar{k}_{on}(1 - E_i(t))\delta_{i0}. \tag{3}$$

A similar equation for $E_i(t)$ is obtained for a multi-occupancy model but with a different pre-factor in the Kronecker delta term[71] i.e. the steady-state distribution of both models have the same form. This is most easily described in the continuum limit ($h \to 0$, $d \to \infty$, $\bar{k}_{on} \to \infty$ keeping $D = dh^2$ and $k_{on} = \bar{k}_{on}h$ fixed) in which we obtain

$$\frac{\partial E(x,t)}{\partial t} = D\frac{\partial^2 E(x,t)}{\partial t^2} - k_{off}E(x,t) + k_{on}(1 - E(x,t))\delta(x). \tag{4}$$

The steady-state solution of this equation is

$$E(x) = \frac{k_{on}}{2\frac{D}{\lambda} + k_{on}}e^{-|x|/\lambda} \tag{5}$$

where $\lambda = \sqrt{D/k_{off}}$ is the associated diffusive lengthscale.

**Fit to ChIP-Seq profile.** Before fitting to the experimental ChIP-seq profile we first binned the profile at 20 bp resolution to match the simulations. We then fit to the right side of the right most peak of the experimental profile (Fig. 3d) to an exponential $y = ae^{x/\lambda}$ as expected from the analysis above. We use the MATLAB *fit* function to fit for the length scale parameter $\lambda$, for which we find $\lambda = 710$ bp. The analysis above shows that $\lambda = \sqrt{\frac{D}{k_{off}}}$ and we confirm this numerically. We can therefore use the estimate for $k_{off}$ obtained from the FRAP experiment to obtain $D = 5600$ bp$^2$s$^{-1}$ = $6.1 \times 10^{-4}$ µm$^2$s$^{-1}$. Note that this value of $D$ does not account for any roadblocks beyond the effect of sliding ParB dimers on each other. The remaining parameter of the sliding model is the overall factor of the loading rates, $k_{on}$. This is determined by finding the best fit of the stochastic model to the entire ParB binding profile as determined by the mean square error between the ChIP-Seq profile and the steady-state profile obtained from the simulations (Supplementary Fig. 3c). Both profiles are normalised to the same area under the curve before the mean square error is calculated.

**Roadblock simulations.** For the roadblock simulations we used the same framework but with a single parS site and choose a high loading rate $k_{on} = 100$ such that this site is occupied the majority of the time. We use the values of $D$ and $k_{off}$ as above. A roadblock is implemented as another particle that can bind and unbind to and from a specific site 25 lattice sites (500 bp) away from the parS site.

To explore the effect of the roadblock we either (1) fix the unbinding rate $k_{R,off}$ such that the residence half-time ($\tau = \log(2)/k_{R,off}$) of the roadblock remains constant and then vary the binding rate $k_{R,on}$ to vary the occupancy of the roadblock, or (2) fix the occupancy ($k_{R,on}/(k_{R,on} + k_{R,off})$) of the roadblock and vary both $k_{R,on}$ and $k_{R,off}$ by the same factor.

## Coupled bridging and sliding model

In the coupled simulations, the bridging probability is dependent on the actual locations of sliding ParB dimers on the polymer. Bridged ParB dimers do not diffuse along the DNA due to the topological constraints of being bound to distal DNA regions. Therefore, bridged dimers act as roadblocks for the unbridged dimers preventing them from sliding past. We keep the polymer at the same length as previously (500 monomers) however we add a further 250 lattice sites to each end over which sliding is also allowed. This provides sufficient length to account for rare far-diffusing ParB dimers before they dissociate.

Parameters are taken from those found in the previous simulations: $p$ is either 40 or $4 \times 10^3$ s$^{-1}$, for the structured and globular regimes respectively, and the ParB unbinding rate $k_{ub} = \frac{\log(2)}{64}$ s$^{-1}$, and the overall ParB loading rate $k_{on} = 200 \cdot k_{off}$s$^{-1}$. The diffusion coefficient from the sliding model cannot be used due to the roadblock effect of the bridges. It is therefore chosen along with the bridging rate $k_b$ to best fit the ChIP-seq profile while at the same time resulting in the observed level of condensation (78 nm ParB radius). Increasing the diffusion coefficient beyond a certain point has no effect due to sliding ParB dimers reaching steady-state between bridges.

The simulation is run until steady state has been reached, as determined by the volume occupied by the polymer, before a snapshot is taken. For each parameter set tested 1000 simulations are ran, each with a different initial conformation.

### Comparison to previous models

The closest model to the current work is the spreading and bridging model[20]. This equilibrium model showed how ParB could compact the centromeric region through a combination of bridging and nearest-neighbour interactions. At the time it was not known that ParB is a DNA clamp and the model therefore relied on the binding of ParB to non-specific DNA. While the model nominally accounted for the stiffness of the DNA, the polymer model used allowed only 0° or 90° bond angles and so was unlikely to fully capture the interplay between ParB bridges and stiffness that we observe in our model. The nucleation and caging model[11,21] can not be directly compared with ours as it does not model ParB explicitly but rather treats it as a fixed spatial distribution centred on the parS site. This was proposed to be a result of ParB self-assembly due to self and non-specific DNA interactions together with nucleation at the parS site (later described as liquid-liquid phase separation[43]). Unlike our model, ParB had no effect on the DNA (bridges were not considered), which was modelled as a Gaussian polymer. While this model required an unphysically short persistence length to explain the ChIP-Seq binding profile, this was subsequently resolved by including the effect of DNA supercoiling[22].

### Fluorescence recovery after photobleaching (FRAP)

*C. crescentus* strain MT174 (parB::egfp-parB)[72] was grown in M2G minimal medium[73] at 28 °C and 220 rpm for 36 h to an OD600 of ~0.6. Cells were spotted on pads made of 1% (w/v) agarose in M2G medium. Images were taken with a Zeiss Axio Imager.Z1 microscope equipped with a Zeiss Plan Apochromat 100x/1.46 Oil DIC objective and a pco.edge 4.2 sCMOS camera (PCO). An X-Cite 120PC metal halide light source (EXFO, Canada) and an ET-EGFP filter cube (Chroma, USA) were used for fluorescence detection. FRAP analysis was conducted by bleaching single EGFP-ParB foci using a 488 nm-solid state laser and a 2D-VisiFRAP multi-point FRAP module (Visitron Systems, Germany),

with 2-ms pulses/pixel at 20% laser power. After the acquisition of a prebleach image and application of a laser pulse, 16 images were recorded at 20 s intervals with VisiView 4.0.0.14 (Visitron Systems). For each time point, the average fluorescence intensities of equally sized regions containing the bleached focus, the non-bleached focus, the cell background and a reference region of the agarose pad were determined, using Fiji 1.49[74]. After background correction, normalisation and averaging of the focus intensities, the recovery half-time was calculated by fitting the data as described below.

**Analysis.** To calculate the residence time of ParB dimers from the photobleaching we perform a simple manipulation of the data. Following the standard calculation used in[43] the FRAP experiments can be described by a simple kinetic model for the ParB proteins in the partition complex and the ParB in the rest of the cytoplasm. Considering $B_1(t)$ and $B_2(t)$ as the average number of ParB proteins in each partition complex after photobleaching, $B_{tot}$ the total number of ParB dimers, and $k_{in}$ and $k_{out}$ the rate to enter and exit the partition complexes respectively, the system can be written as:

$$\frac{dB_1(t)}{dt} = k_{in}B_{tot} - (k_{in} + k_{out})B_1(t) - k_{in}B_2(t), \qquad (6)$$

$$\frac{dB_2(t)}{dt} = k_{in}B_{tot} - (k_{in} + k_{out})B_2(t) - k_{in}B_1(t). \qquad (7)$$

In order to fit to the data more easily we consider the sum and difference, $B_\pm = B_1(t) \pm B_2(t)$:

$$\frac{dB_+(t)}{dt} = 2k_{in}B_{tot} - (2k_{in} + k_{out})B_+(t) \qquad (8)$$

$$\frac{dB_-(t)}{dt} = -k_{out}B_-(t). \qquad (9)$$

The general solution to these equations is given by:

$$S_+ = 2S_\infty - 2\left[S_\infty - \frac{1}{2}S_+(0)\right]e^{-(2k_{in}+k_{out})t} \qquad (10)$$

$$B_-(t) = -B_-(0)e^{-k_{out}t}. \qquad (11)$$

A simple exponential fit of our data to the difference curve (Supplementary Fig. 3b) finds $k_{out} = 0.011\ \text{s}^{-1}$, or a half-time in the focus of 64 s and $B_-(0) = 0.91$. Then fitting to the sum, taking $B_\infty$ to be equal to 0.62, we find $B_+(0) = 1.06$ and $k_{in} = 0.0035\ \text{s}^{-1}$. Using these fitted values we can plot $B_1(t)$ and $B_2(t)$ in Fig. 3c using the simple transformation $S_1 = \frac{1}{2}(S_+ + S_-)$ and $S_2 = \frac{1}{2}(S_+ - S_-)$.

**Reporting summary**

Further information on research design is available in the Nature Portfolio Reporting Summary linked to this article.

## Data availability

The data sets generated from the polymer model can be found at https://gitlab.gwdg.de/murray-group/kBFM/-/tree/bridging_prob and the data sets generated from the coupled bridging and sliding model at https://gitlab.gwdg.de/murray-group/kBFM/-/tree/coupled. The accession number for the ChIP-seq data is GSE100233[12]. Source data are provided with this paper.

## Code availability

Code for the polymer simulation is available at https://gitlab.gwdg.de/murray-group/kBFM/-/tree/bridging_prob. Code for the sliding simulation is available at https://gitlab.gwdg.de/murray-group/Caulobacter_

ParABS/-/tree/Caulo_20bp. Code for the coupled polymer and sliding simulation is available at https://gitlab.gwdg.de/murray-group/kBFM/-/tree/coupled.

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

## Acknowledgements
We thank Jean–Charles Walter (Montpellier) for comments on the manuscript. This work was funded by the Max Planck Society.

## Author contributions
S.M.M. conceived the project. S.M.M. and L.C. performed the computational modelling and analysis. M.T. and L.S. performed the FRAP experiments. S.M.M. and L.C. wrote the manuscript.

## Funding

## Competing interests
The authors declare no competing interests.
