## [Peer Review File · Nature Communications]

REVIEWER COMMENTS

Reviewer #1 (Remarks to the Author):

Murray and his group focused on an interesting problem of bacterial genome partition, which may have important implications on the fundamental principles of genome partitions in general. Bacterial genome partitioning is controlled by the ParABS system in many model systems. The formation of the partition complex—emerging from the multi-valent interactions between ParB dimers and the ParS-containing centromere-like regime of the genome—underlies the fidelity of genome partition. The authors presented a computational model to explore how the partition complex structures of the ParABS system emerge via the so-called sliding and bridging of ParB dimers. The authors claimed that the transient ParB bridges without hindering the ParB sliding along the DNA are essential for both the condensation of the nucleoprotein complex and the multi-peak ParB binding profile around the Ori regime of the bacterial genome. The manuscript is clear and well-written and touches upon a significant question.

However, I am not convinced that their work presents a sufficient conceptual leap that warrants the novelty for publishing in Nature Communications. This is a highly contested field: Many theoretical models are already presented, which have detailed descriptions of how the spreading, bridging, and/or caging processes of ParB along the DNA promotes the formation of partition complexes (e.g., the ref # 11, 20-22, 41, 54). Some of these published works even combined extensive experimental testing and validation of the respective theoretical model (e.g., ref #11 & 54). Additionally, a core model ingredient (ParB sliding) in the current manuscript was already presented in the authors' previous paper (e.g., ref #13). Given that so much has been done already, it is difficult to appreciate what the new finding with significance of this work is, even if the theoretical models and treatments and the conclusions are on a solid base, the issues of which I will comment in the technical suggestions below.

1. The authors first considered two simplified models studying the bridging and sliding processes separately. In the first simplified model, the authors studied how a ParB distribution on the DNA can generate various partition complex structures through the ParB bridging process, while assuming a fixed ParB distribution along the DNA by mapping the experimentally observed ParB genomic distribution. In the second simplified model, the authors considered how ParB sliding along the DNA—without considering the bridging effect—can reproduce the experimentally observed multi-peaked CHIP-seq profile. The authors fitted these simple models to the experimental data to determine the effective diffusion coefficient and the effective dissociation rate of ParB along, and from, the DNA, respectively.

The problem here is that in the partition complex formation, the processes of ParB sliding and the ParB bridging are coupled together. Fixing one process and use the other to fit the experimental data may

introduce large error in determining the model parameters. For instance, in their “Semi-flexible polymer model of ParB bridging”, what is the physical basis for the assumption that ParB spreading is independent of the 3D structure of DNA? The authors should use their third model that combines the sliding and bridging processes to fit the data to determine the model parameters.

2. For the ParB sliding, it is confusing why the DNA is modeled as a 1D lattice. This is because DNA is a 1D manifold (x-axis) in a 3D space (x-y-z); this way, the DNA-bound proteins may still “pass” each other if they are in the different y-and z-coordinates. Additionally, ParB dimers may necessarily only undergo sliding along the DNA, but they can hop along the DNA and diffuse around in 3D, which may be important considering the DNA have conformational changes in 3D. In the latter case, whatever the ParB bridging effects are, they may not necessarily hinder the hopping process.

3. What is the intrinsic relaxation timescale of the DNA conformational change in the semi-flexible polymer model? What determines this timescale? And how does it compare with the kinetic rates of ParB sliding and bridging process? If this timescale is not experimentally measured, then how does its variation impact the physical picture of partition complex formation and the model conclusion? Furthermore, in a coarse-grained model, the choice of simulation time step and duration may have adverse effects. Therefore, the authors should show the timestep and duration in the model simulation for the sliding-and-bridging coupled model, and how these choices affect the results.

4. There are typos in the manuscript. For instance, the last sentence in the 4th or 5th paragraph on p7 is incomplete.

5. It is important for the authors to parse out 1) the exact differences between their model and others (e.g., those in ref #11, 20, 41 etc), and 2) propose experimental testing to distinguish these models.

To sum up, I'd suggest that a more specialized journal may be appropriate for this work, after the authors addressing the above points.

Reviewer #2 (Remarks to the Author):

in this ms, Lara Connolley and collaborators developed a polymer physics frameworks to address the problem of the formation of the ParBS partition complex. In particular, they investigate whether the

sliding and bridging properties of ParB, which have been demonstrated experimentally, can reproduce both the phenomenology of the ParB cluster and of its binding profile along DNA in the context of the chromosome of *C. crescentus*. They then use their framework to predict potential DNA conformations that could occur during the formation of the partition complex.

The work by Connoley and collaborators provides novel interesting insights into the problem of the formation of the partition complex, providing in particular a unified picture of sliding and bridging. It is expected to be of interest not only to the community of scientists working on this topic but, more generally, to the community of scientists working on the problem of DNA structuring in bacteria.

I have two main criticisms that the authors should consider to improve the relevance of their findings as well as the soundness of their computational framework.

1. First, the phase diagram should be hard to apprehend for a general audience without going into the details of the model (as I had to do). In particular, the authors use a pure kinetic terminology (e.g. the binding lifetime along the y-axis) and explain the structuring phenomenology also in terms of kinetic terms, only. However, the problem studied in the first part is a thermodynamic problem at equilibrium (the problem of the collapsed of a polymer in the presence of bridging proteins). This would therefore make sense to use appropriate adimensional parameters as commonly used in the field of the structuring properties of chromosome.

From a descriptive viewpoint, I therefore encourage the authors to adopt the classical description used for discussing the problem of the collapse of a polymer in the presence of bridging proteins: the concentration of the proteins versus the bending energy of the proteins (see e.g. figure 1B of [45]). More precisely:

- the “concentration” should be something like $k_b \times b^2 / (k_{ub} + k_b)$ where b is the average ParB occupancy. Note, then, that this roughly corresponds to the x-axis of Fig. 1C but with the additional term b^2 . The latter should then rescale the whole axis so that values are on the order 1 (what one would expect for a relevant description of the thermodynamic properties of the system).

- the “bending energies” in the model correspond to $-k_B T \log(k_{ub} \times \tau)$, where τ is the time unit used in the simulation. Note then with respect to the (non-standard) terminology used by the authors, τ should be on the order of τ_0/p (where τ_0 is fixed), while k_{ub} on the order of $1/\lambda$ such that this energy should be on the order of $k_B T \log(p\lambda/\tau_0)$, i.e., $k_B T \log(\theta/\tau_0)$ — the latter reflects the link between the thermodynamics and the kinetic properties. The discussion on the timescales (p , λ , k_{ub} , θ) could also be simplified.

From an explanatory viewpoint, the paragraph “We propose that these two...” and the section “Short-lived PARB bridging” should at least be completed by a thermodynamic discussion. The authors could inspire from the discussion provided in [39] (see in particular explanation in Fig. 6 of that article). Note also that in [39], authors discuss the geometry of the collapsed state in terms of the ratio of the size of the binders with respect to the stiffness of the polymer. In particular, for high ratio, the binders aggregate instead of forming hairpins. This could be added to improve the discussion about the expected effect of fitness.

2. Second, and more importantly, the method used to simulate the combination of sliding and bridging appears unjustified to me. Providing a physically justified method is all the more important that kinetic details, here, play a priori an important role since the problem involves diffusion and, hence, is not at equilibrium. I think that the proper simulation of 1D diffusion and 3d folding is required, all the more that the fluctuating bond model provides a reasonable kinetic description of the dynamics of a polymer.

Along this line, an important aspect that is not discussed in the ms concerns the fact that the full model (binding + sliding) is actually an out-of-equilibrium system. In particular, there is a tacit continuous input of ParB proteins that should be discussed. Indeed, once ParB unbinds from the DNA, it could a priori freely diffuse in the cell. In this situation, one might ask how it “re-targets” parS in a reasonable time (see von Hippel work for the problem of 3D targeting of a protein). For instance, the work in [22] considered this aspect and actually required ParB to be produced close to the ParS to solve the paradox. This seems to be a necessary condition as well here.

3. In addition, I provide a list of minor comments that authors should consider in order to improve the clarity as well as the generality of the ms:

- Introduction: “the previous models need to be reevaluated” => I understand that model in [13] already includes sliding. It looks to me that the novelty here is that the authors provide a unified framework (with respect to previous models) that combines both sliding and bridging. In particular, as explained by the authors, this may provide a framework to make the connection between chromosome data and F-plasmid data.

- At the beginning of the results part, it would be useful to provide information such as: the size of the molecule (10 kb), the spatial resolution (2.6 nm) and also that the molecule is linear

- the authors mention that BFM reproduces Rouse polymer dynamics as an interesting feature of the model. This might be too much of a jargon for e.g. biologists. For instance, it is not relevant for the

discussion of the phase diagram (see main concern 1 above). It is actually relevant for the full model (ibid) because it provides a realistic dynamics for a polymer chain.

- it is not clear how a coil is defined in the ms (I understood that it corresponded to the situation of less than 20 bridges, which look arbitrary). In polymer physics, it is usually defined by looking at the point where the radius of gyration collapse (let's say by a factor 2 with respect to the swollen state)

- it would be useful to mention whether the 78 nm size of the ParB cluster of *C. crescentus* was measured using super-resolution microscopy since this type of experiments have led to a reestimation (by a factor 2) of the cluster diameter in the case of the F-plasmid (which is also the reason of the work in [22]).

- first sentence of the section "ParB sliding model can reproduce the multi-peaked profile" : In the last sections => In the previous sections

- it would be useful to mention that the diffusion constant of $610 \text{ nm}^2 \text{ s}^{-1}$ is much smaller than that reported for the 1D diffusion of transcription factor (LacI, see Xie's group work). Also, in the full model, the value of the diffusion constant that is used to recover the ParB profile should be given.

- last sentence of the paragraph "Coupled simulations of sliding and bridging": the authors use the word "equilibrium". This should be replaced by "stationary" as the system is out-of-equilibrium (see main concern 2 above).

- end of the results section: problem of sentences that were cut into two parts.

- discussion: I do not agree with the sentence "Indeed, in the absence of a specific mechanism, DNA hairpins are unlikely to form given the intrinsic stiffness of the DNA". DNA supercoiling, which is not considered here, is an important feature of bacterial chromosomes that is known to induce the formation of plectonemic structures, whose topology is exactly that of an hairpin. The discussion about DNA supercoiling is actually reduced to a single sentence "Supercoiling, which is not accounted for in our model, may also contribute". This is a bit frustrating, especially for a readership that should be interested in the structuring of bacterial chromosomes. For instance, helical structures found by the authors are unlikely to occur in this context.

Reviewer #3 (Remarks to the Author):

This manuscript reports a study of how the ParB-parS partition complex assembles in bacteria. This complex is a major actor in genome (both chromosomes and plasmids) segregation. Homologous systems also position other complexes involved in chemotaxis, mobility etc., emerging as a global mechanism of cell patterning in bacteria, reinforcing the crucial importance of its understanding. The literature is currently rich, particularly with the recent discovery of a new and original switch in the ParB protein, using CTP binding/hydrolysis to close a DNA clamp at the parS sites, then open it after sliding for some distance. This has further strengthened the interest for ParB-parS assembly and how these new data can be incorporated into the previous models and fit the existing data is still a matter of debate. The current manuscript is a nice add on to this field since it provides a model integrating the known activities of ParB and fitting the data in the case of the ParB-parS complex of the model bacterium *C. crescentus*.

As a pure biologist, the technical aspect of the calculations is far beyond my knowledge. I have one major concern, which may result from the writing and organization of the manuscript or from a real scientific flaw. Along the text, the authors calculate parameters to feed their models from ChIP-seq data, assuming their hypotheses are true (i.e. the repartition of ParB dimers is due to sliding from parS sites, allowing calculation from the ChIP-seq data of e.g. the diffusion coefficient and the dissociation rate). By doing so, parameters are found that fit quite well the data, showing that the starting hypothesis is possible. This however does not rule out other hypotheses. For example, the ParB-ParB recruitment, when added into the model, makes it fitting less well. Have (or can) this mechanism been taken into account to recalculate the parameters ? Could a better fit be reached with ParB-ParB recruitment ? The authors should aim to make this clearer for the general reader or leave them with an impression of instruction in charge.

Abstract : Is it useful to talk about speculation in the abstract ? There is no data supporting a role in SMC loading here, which is a pure hypothesis, thus welcome only in the discussion.

Last paragraph of page 2 : the rationale about the degree of movement which may be different in the two regimes is difficult to follow. I have no solution to propose but the authors should attempt a clearer explanation.

Page 3 'short-lived ParB bridging leads to ...' : It is stated here that weak bridging leads to a regime with hairpins and helices. DNA supercoiling should have an important effect on the formation? dynamics? of these structures ? Can this be ignored ? It should at least be discussed better than just one sentence in the discussion section.

Fig1 : Is the red dot a parS site in panel A ? If so, why is there a single one here ? It would help to state early in the text and in the legend that two of the 5 parS are too close to be clearly distinguished in the figs.

Fig2 : What are the dotted rectangles in panel A ? Does 'mons' state for 'monomers' in the Y-axis ? The shading is difficult to see in panel B.

Page 5 first paragraph : It seems to me that other authors have reported CHIP-seq patterns from a chromosome with multiple parS: Vibrio, others ? This may deserve to be cited here. Could the data be used to test the models presented here ?

Third paragraph : FRAP data have been obtained with the F system. This should be cited and possible differences discussed.

Fourth paragraph, second sentence 'previous measurements' : please provide a reference for this.

Page 6 first paragraph of 'roadblocks' : the juggling between lifetime and % of occupancy is not obvious here for the non-specialist and only becomes clear when reading the figure and legend, please explain better.

Page 7 top : parameters (diffusion coefficient) need to be (and are) changed here to restore the observed distribution of ParB on the DNA. Why not later when the ParB-ParB recruitment is implemented ?

'This suggests that sliding of ParB pushes the system towards the structure regime' : can the authors provide an (intuitive) explanation for this ?

Last paragraph of the result section : There has been a problem with the text here (part is missing), please correct.

Reviewer #1:

Murray and his group focused on an interesting problem of bacterial genome partition, which may have important implications on the fundamental principles of genome partitions in general. Bacterial genome partitioning is controlled by the ParABS system in many model systems. The formation of the partition complex – emerging from the multi-valent interactions between ParB dimers and the ParS-containing centromere-like regime of the genome – underlies the fidelity of genome partition. The authors presented a computational model to explore how the partition complex structures of the ParABS system emerge via the so-called sliding and bridging of ParB dimers. The authors claimed that the transient ParB bridges without hindering the ParB sliding along the DNA are essential for both the condensation of the nucleoprotein complex and the multi-peak ParB binding profile around the Ori regime of the bacterial genome. The manuscript is clear and well-written and touches upon a significant question.

However, I am not convinced that their work presents a sufficient conceptual leap that warrants the novelty for publishing in Nature Communications. This is a highly contested field: Many theoretical models are already presented, which have detailed descriptions of how the spreading, bridging, and/or caging processes of ParB along the DNA promotes the formation of partition complexes (e.g., the ref # 11, 20-22, 41, 54). Some of these published works even combined extensive experimental testing and validation of the respective theoretical model (e.g., ref #11 & 54). Additionally, a core model ingredient (ParB sliding) in the current manuscript was already presented in the authors' previous paper (e.g., ref #13). Given that so much has been done already, it is difficult to appreciate what the new finding with significance of this work is, even if the theoretical models and treatments and the conclusions are on a solid base, the issues of which I will comment in the technical suggestions below.

Whilst it is true that much previous work has been done on modelling partition complex formation, the understanding of how ParB dimers can organise and compact the DNA is still limited. There are not multiple competing models as this reviewer makes out. For example the model of ref. 11 refuted the model of ref 20, at least for F plasmid, and ref 22 states that the previous model of the same authors (ref. 11, 21) is “difficult to justify on physical grounds, namely, a very small DNA persistence length of ~10 bp” amongst other reasons. Thus the models have evolved over time as more knowledge of the system has been gained.

Additionally, only the model of ref 20 has previously considered DNA bridging by ParB explicitly but without a proper representation of the stiffness of the DNA. In other studies bridging was either not included with ParB itself treated implicitly and having no effect on the DNA (refs 11, 21, 22) or only in a mean field approximation with associated assumptions (ref. 23). Thus, the comment ignores the significant part and new results of this work on the effect of ParB-induced DNA bridging.

Most importantly, it has recently been definitively proven that ParB dimers are DNA clamps that can slide along the DNA (for the ParBs of *Caulobacter crescentus*, *Myxococcus xanthus*, *Bacillus subtilis* and strongly indicated for F plasmid). However, no model of complex formation has yet incorporated this important fact (indeed in the models of ref. 11, 21, 22 ParB is treated implicitly). Sliding also means that the complex is out of equilibrium in that there is a net flux out from the parS sites and detailed balance does not hold. All the models

so far have been equilibrium models. Therefore we do not understand how this reviewer can dismiss the novelty, timeliness and relevance of our study.

Finally, whilst we have previously presented a simple sliding model for ParB spreading, that study did not consider the effect of transient protein roadblocks. Moreover, here we combine bridging and sliding in one unified model. For all these reasons we believe our work is an important contribution to our understanding of how this system functions.

We have added a section to the SI discussing in more detail the difference between the model presented here and previous models.

1. The authors first considered two simplified models studying the bridging and sliding processes separately. In the first simplified model, the authors studied how a ParB distribution on the DNA can generate various partition complex structures through the ParB bridging process, while assuming a fixed ParB distribution along the DNA by mapping the experimentally observed ParB genomic distribution. In the second simplified model, the authors considered how ParB sliding along the DNA – without considering the bridging effect – can reproduce the experimentally observed multi-peaked CHIP-seq profile. The authors fitted these simple models to the experimental data to determine the effective diffusion coefficient and the effective dissociation rate of ParB along, and from, the DNA, respectively.

The problem here is that in the partition complex formation, the processes of ParB sliding and the ParB bridging are coupled together. Fixing one process and use the other to fit the experimental data may introduce large error in determining the model parameters. For instance, in their “Semi-flexible polymer model of ParB bridging”, what is the physical basis for the assumption that ParB spreading is independent of the 3D structure of DNA? The authors should use their third model that combines the sliding and bridging processes to fit the data to determine the model parameters.

We agree that presenting either simply the ParB bridging or the ParB sliding separately is a simplification of the biological system. However, this simplification allows us to probe in more detail the effects of different parameters on the system without being restrained by the computational limits of simulating the coupled system. We then finally present the coupled system to demonstrate that the two processes can indeed both co-exist. We do not claim to have found a unique parameter fitting and make no claims regarding the relevance of specific parameter values. Our point is simply that the coupled models can reproduce the observed ChIP-Seq profile and a compacted partition complex. We have amended the text to make clearer the final parameters determined by our coupled model.

In particular, the assumption that ParB spreading is independent of the 3D structure of DNA is based on the fact that ParB dimers can form clamps which bind at the *parS* sites and subsequently slide along the DNA. Notwithstanding the effect of roadblocks including due to ParB bridges, which as stated we explicitly ignore here, we have no reason to believe that this sliding process would depend on the 3D conformation of the DNA, especially given the small size of the ParB footprint compared to the persistence length.

We would also contrast our assumption to that made in the models of ref 11, 21 and 22, which assume that the spatial distribution of ParB around the parS site and the conformation of the DNA are independent of one another, thereby ignoring the effect of ParB bridging. Simplifying assumptions are a necessary component of mathematical modelling and can still result in useful models.

2. For the ParB sliding, it is confusing why the DNA is modeled as a 1D lattice. This is because DNA is a 1D manifold (x-axis) in a 3D space (x-y-z); this way, the DNA-bound proteins may still “pass” each other if they are in the different y-and z-coordinates.

We do not understand the issue here. Our sliding simulation models the DNA strand itself as a 1D and single occupancy lattice and we are interested in the position of the ParB dimers along the strand. ParB dimers entrap and slide along the DNA and there is no reason to believe, based on the structures and the experimental evidence thus far, that they could pass through each other.

Additionally, ParB dimers may necessarily only undergo sliding along the DNA, but they can hop along the DNA and diffuse around in 3D, which may be important considering the DNA have conformational changes in 3D. In the latter case, whatever the ParB bridging effects are, they may not necessarily hinder the hopping process.

In our model we assume that unclamped ParB is well-mixed in the cytosol. This is justified by its weak affinity for DNA and the rapid diffusion of small proteins in the cytosol on the timescale of ParB turnover (~60s). The measured diffusion coefficient of cytosolic ParB_F of 0.1-1 $\mu\text{m}^2/\text{s}$ is consistent with this. Even if there was a local increase in the cytosolic population around the parS site, there is no reason to expect it would fundamentally affect our results. The only difference would be that a lower loading rate of ParB onto the parS sites would suffice to load the same amount of ParB.

3. What is the intrinsic relaxation timescale of the DNA conformational change in the semi-flexible polymer model? What determines this timescale? And how does it compare with the kinetic rates of ParB sliding and bridging process? If this timescale is not experimentally measured, then how does its variation impact the physical picture of partition complex formation and the model conclusion?

All our simulations use the kinetic Monte Carlo / Gillespie algorithm method. The intrinsic relaxation timescale of the DNA polymer in our model is determined by the overall move attempt rate parameter, p . Whilst this variable is not experimentally known the variation of it in relation to the ParB bridge lifetime is precisely what we probe in the first section of the paper where we show that different ParB bridge lifetimes (in units of $1/p$) result in distinctly different conformations. While this was stated in the legend to Figure 1 and in the methods, we have now made this clearer.

Furthermore, in a coarse-grained model, the choice of simulation time step and duration may have adverse effects. Therefore, the authors should show the timestep and duration in the model simulation for the sliding-and-bridging coupled model, and how these choices affect the results.

Our simulations are kinetic so there is no fixed time step to report. This is still the case for the coupled simulations, the time step Δt in Fig. 4a is the dynamic time it took for the last bridging or unbridging event to occur. The only difference between the coupled simulations and a true kinetic implementation is that new bridging events occur based on the ParB locations at the end of previous iteration of the sliding simulation. This is a reasonable simplification as long as the sliding ParBs equilibrate faster the bridging timescale, which we expect to be the case given the ~ 1 min turnover. In any case, in response to Reviewer 2, we have now provided a true kinetic coupling of bridging and sliding. Our results are unchanged.

4. There are typos in the manuscript. For instance, the last sentence in the 4th or 5th paragraph on p7 is incomplete.

This was a very unfortunate error and has been corrected.

5. It is important for the authors to parse out 1) the exact differences between their model and others (e.g., those in ref #11, 20, 41 etc), and 2) propose experimental testing to distinguish these models.

We now explain the differences between the various models in an additional methods section. As described above, the only previous model including the DNA polymer and the ParB explicitly (not mean field) is that of ref. 20. However, we do not believe it is our responsibility to propose experimental tests of previous models. With regard to our own model, we clarify potential experimental tests of our prediction of DNA hairpin structures (which no other model predicts). We discuss the possibility to see them using atomic force microscopy in addition to the already stated future possibility that they may be measurable in vivo in the future with improvements to HiC (chromatin capture).

To sum up, I'd suggest that a more specialized journal may be appropriate for this work, after the authors addressing the above points.

We disagree with this statement. As described above we believe that our model presents a significant step forward in understanding partition complex formation. We present for the first time a unified description of a sliding and bridging model taking into account recent experimental results. We believe this work is likely to be of interest to researchers, both experimental and theoretical studying DNA, its organisation and its segregation. Therefore we believe that Nature Communications is precisely the right home for our work.

Reviewer #2:

in this ms, Lara Connolley and collaborators developed a polymer physics frameworks to address the problem of the formation of the ParBS partition complex. In particular, they investigate whether the sliding and bridging properties of ParB, which have been demonstrated experimentally, can reproduce both the phenomenology of the ParB cluster and of its binding profile along DNA in the context of the chromosome of *C. crescentus*. They then use their framework to predict potential DNA conformations that could occur during the formation of the partition complex.

The work by Connoley and collaborators provides novel interesting insights into the problem of the formation of the partition complex, providing in particular a unified picture of sliding and bridging. It is expected to be of interest not only to the community of scientists working on this topic but, more generally, to the community of scientists working on the problem of DNA structuring in bacteria.

I have two main criticisms that the authors should consider to improve the relevance of their findings as well as the soundness of their computational framework.

1. First, the phase diagram should be hard to apprehend for a general audience without going into the details of the model (as I had to do). In particular, the authors use a pure kinetic terminology (e.g. the binding lifetime along the y-axis) and explain the structuring phenomenology also in terms of kinetic terms, only. However, the problem studied in the first part is a thermodynamic problem at equilibrium (the problem of the collapsed of a polymer in the presence of bridging proteins). This would therefore make sense to use appropriate adimensional parameters as commonly used in the field of the structuring properties of chromosome.

We note that the parameters on the phase diagram are non-dimensional. $\theta = p/k_{ub}$ the bridge lifetime in units of the reciprocal of the move rate. This was stated in legend but we agree it was not clear from the axis label.

From a descriptive viewpoint, I therefore encourage the authors to adopt the classical description used for discussing the problem of the collapse of a polymer in the presence of bridging proteins: the concentration of the proteins versus the bending energy of the proteins (see e.g. figure 1B of [45]). More precisely:

- the “concentration” should be something like $k_b \times b^2 / (k_{ub} + k_b)$ where b is the average ParB occupancy. Note, then, that this roughly corresponds to the x-axis of Fig. 1C but with the additional term b^2 . The latter should then rescale the whole axis so that values are on the order 1 (what one would expect for a relevant description of the thermodynamic properties of the system).

- the “bending energies” in the model correspond to $-k_B T \log(k_{ub} \times \tau)$, where τ is the time unit used in the simulation. Note then with respect to the (non-standard) terminology used by the authors, τ should be on the order of τ_0/p (where τ_0 is fixed), while k_{ub} on the order of $1/\lambda$ such that this energy should be on the order of $k_B T \log(p\lambda/\tau_0)$, i.e., $k_B T \log(\theta/\tau_0)$ — the latter reflects the link between the thermodynamics and the kinetic properties.

The discussion on the timescales (p , λ , k_{ub} , θ) could also be simplified.

The bridging energy between monomers i and j is $\log(k_b/k_{ub} B_i B_j)$ and along the lines of the reviewer’s suggestion, we now use $k_b/k_{ub} \langle B_i B_j \rangle$ as the x axis of the phase diagram i.e. the mean exponential of the energy. We do not have a clear ‘concentration’ to use here as ParB is not modelled explicitly in these first simulations. We considered the suggestion of the reviewer to use $\langle B_i \rangle^2 \times k_b / (k_{ub} + k_b)$ as a measure of the concentration of bridges. However, this does not account for the conformation of the polymer (e.g. the two indicated points in the phase diagram have the same number of bridges but different k_b/k_{ub}).

The y-axis remains as p/k_{ub} , the lifetime of a bridge in units of $1/p$ but we now make clear via the new diagram in Fig. 1 showing the energy landscape of bridging and unbridging, its thermodynamic interpretation as the (exponential of the) activation energy for bridging.

Leaving this description in terms of kinetic parameters also makes the transition to the coupled model easier. As pointed out by the reviewer, our coupled model is out-of-equilibrium thus we necessarily must utilise a kinetic description. Describing the uncoupled, equilibrium, polymer model in a different manner would make the transition to the coupled case awkward in our opinion. Furthermore it is not unusual to use kinetic terminology to describe bridging. Such a description was also used by Bohn and Heermann (ref 59), despite being primarily interested in equilibrium properties.

From an explanatory viewpoint, the paragraph “We propose that these two...” and the section “Short-lived PARB bridging” should at least be completed by a thermodynamic discussion. The authors could inspire from the discussion provided in [39] (see in particular explanation in Fig. 6 of that article). Note also that in [39], authors discuss the geometry of the collapsed state in terms of the ratio of the size of the binders with respect to the stiffness of the polymer. In particular, for high ratio, the binders aggregate instead of forming hairpins. This could be added to improve the discussion about the expected effect of fitness.

We have modified the explanation describing how the two different compact polymer regimes arise to add in a more thermodynamic discussion. However, we believe that our observations are due to kinetic effects as in the context of Fig. 6 of ref [39] so kinetics naturally appear in our explanation.

We have also added a comment in the discussion regarding binder size vs stiffness and cited ref. 39.

2. Second, and more importantly, the method used to simulate the combination of sliding and bridging appears unjustified to me. Providing a physically justified method is all the more important that kinetic details, here, play a priori an important role since the problem involves diffusion and, hence, is not at equilibrium. I think that the proper simulation of 1D diffusion and 3d folding is required, all the more that the fluctuating bond model provides a reasonable kinetic description of the dynamics of a polymer.

The only aspect of the approach that differed from a true kinetic implementation was that the formation of a bridge used the position of sliding ParBs at a previous time point (the previous iteration). This was a valid approximation as long as sliding ParB equilibrates faster than the bridging timescale. In any case, we have now properly integrated ParB sliding into the bridging polymer into a single combined kinetic implementation. Our results remain unchanged.

Along this line, an important aspect that is not discussed in the ms concerns the fact that the full model (binding + sliding) is actually an out-of-equilibrium system. In particular, there is a tacit continuous input of ParB proteins that should be discussed. Indeed, once ParB unbinds from the DNA, it could a priori freely diffuse in the cell. In this situation, one might ask how it

“re-targets” parS in a reasonable time (see von Hippel work for the problem of 3D targeting of a protein). For instance, the work in [22] considered this aspect and actually required ParB to be produced close to the ParS to solve the paradox. This seems to be a necessary condition as well here.

We have updated the text to better acknowledge that the coupled model is out-of-equilibrium, including raising the point in the introduction. This is an important physical consequence of the discovery of ParB sliding. In our sliding and coupled simulations, we assume that ParB in the cytosol is well-mixed. This assumption was largely for convenience. However, as we pointed out in our response to reviewer one, the ~60s timescale of ParB turnover likely means that it is the case. On the other hand, a high local concentration of bound ParB around the parS site may very well contribute to the loading rate. In either case, we know ParB can find the parS site sufficiently rapidly to maintain a cluster there (clusters are observed in vivo). How exactly this occurs (facilitated diffusion or otherwise) does not really matter from the point of view of our study and is a question for further work and experiments. Nevertheless, we agree that a discussion of this is appropriate and we now comment on this point in this description of the sliding model.

Regarding the assumption of the cited work [22] that ParB is produced close to the parS sites, we have a hard time believing that for the following reasons: 1) the ribosomes in E. coli are excluded from the nucleoid and are usually found at the poles, whereas the partition complex is known to be within the nucleoid, along the centerline of the cell, at least for F plasmid (Le Gall et al 2016), and 2) most proteins are relatively stable, so that the observed 60s turnover of ParB clusters is unlikely to be explained/replenished by de novo synthesis (our own experiments on F plasmid support this).

3. In addition, I provide a list of minor comments that authors should consider in order to improve the clarity as well as the generality of the ms:

- Introduction: “the previous models need to be reevaluated” => I understand that model in [13] already includes sliding. It looks to me that the novelty here is that the authors provide a unified framework (with respect to previous models) that combines both sliding and bridging. In particular, as explained by the authors, this may provide a framework to make the connection between chromosome data and F-plasmid data.

While our previous paper did introduce a model of sliding, our point here was that models of *partition complex formation* need to be reevaluated. This was not clear and we have updated the introduction. Furthermore, we have also added a section to the SI discussing the differences between our and previous models.

- At the beginning of the results part, it would be useful to provide information such as: the size of the molecule (10 kb), the spatial resolution (2.6 nm) and also that the molecule is linear

These details have been added.

- the authors mention that BFM reproduces Rouse polymer dynamics as an interesting feature of the model. This might be too much of a jargon for e.g. biologists. For instance, it is

not relevant for the discussion of the phase diagram (see main concern 1 above). It is actually relevant for the full model (ibid) because it provides a realistic dynamics for a polymer chain.

We have removed the comment from the main text.

- it is not clear how a coil is defined in the ms (I understood that it corresponded to the situation of less than 20 bridges, which look arbitrary). In polymer physics, it is usually defined by looking at the point where the radius of gyration collapse (let's say by a factor 2 with respect to the swollen state)

The radius of gyration or end-to-end distance etc are not particularly useful here since ParB is very heterogeneously distributed along the polymer. For that reason we use the ParB weighted radius to distinguish between regimes instead. The polymer is described as a coil where there are too few bridges to discern if a conformation is structured or globule-like, the latter distinction being made using the ParB weighted radius. The value of 20 was chosen by observation. Using a two fold decrease in the ParB weighted radius would be too conservative and we would never include the structured regime, which is clearly visible.

We can also not measure the scaling exponent because, apart from the heterogeneity, the polymer is too short given its stiffness to extract the polymer scaling and determine the polymeric regime. This is why we used the term 'globule-like'.

We have changed our description of the coiled state to coil-like.

- it would be useful to mention whether the 78 nm size of the ParB cluster of *C. crescentus* was measured using super-resolution microscopy since this type of experiments have led to a reestimation (by a factor 2) of the cluster diameter in the case of the F-plasmid (which is also the reason of the work in [22]).

The 78nm radius was determined using PALM microscopy, a super-resolution microscopy technique. This has been added to the text.

- first sentence of the section "ParB sliding model can reproduce the multi-peaked profile" :
In the last sections => In the previous sections

Done.

- it would be useful to mention that the diffusion constant of $610 \text{ nm}^2 \text{ s}^{-1}$ is much smaller than that reported for the 1D diffusion of transcription factor (LacI, see Xie's group work). Also, in the full model, the value of the diffusion constant that is used to recover the ParB profile should be given.

We have added the diffusion constant used in the coupled model ($0.0039 \text{ um}^2 \text{ s}^{-1}$) to the text. We note that this is significantly higher than that used in only the sliding model and more inline with a value on the magnitude of $10^{-2} \text{ um}^2 \text{ s}^{-1}$ as has previously been measured experimentally for 1D diffusion of proteins such as LacI (Elf et al, 2007) and also

in vivo measures of ParB [13,42]. Furthermore, it should be noted that this is only a lower bound on the diffusion coefficient (based on a rather arbitrarily chosen bridge lifetime of 1s - a shorter lifetime should result in a corresponding increase in the minimum required diffusion coefficient). Increasing the diffusion coefficient beyond this value has very little effect on the ParB profile recovered since the ParB dimers sliding reach equilibrium between bridging events.

- last sentence of the paragraph "Coupled simulations of sliding and bridging": the authors use the word "equilibrium". This should be replaced by "stationary" as the system is out-of-equilibrium (see main concern 2 above).

An unfortunate error on our part. We were of course aware that the sliding and coupled simulations are out of equilibrium.

- end of the results section: problem of sentences that were cut into two parts.

This issue has been fixed.

- discussion: I do not agree with the sentence "Indeed, in the absence of a specific mechanism, DNA hairpins are unlike to form given the intrinsic stiffness of the DNA". DNA supercoiling, which is not considered here, is an important feature of bacterial chromosomes that is known to induce the formation of plectonemic structures, whose topology is exactly that of an hairpin. The discussion about DNA supercoiling is actually reduced to a single sentence "Supercoiling, which is not accounted for in our model, may also contribute". This is a bit frustrating, especially for a readership that should be interested in the structuring of bacterial chromosomes. For instance, helical structures found by the authors are unlikely to occur in this context.

We thank this reviewer for pointing out that this sentence is indeed incorrect. The hairpin structures do look similar to plectonemes. We now make this clear and that they should be preferred over solenoidal/helical arrangements. We also expect that supercoiling and plectonemes, to the degree that they are relevant on this 1-2kb scale, would only promote the formation of ParB-DNA hairpins.

Reviewer #3:

This manuscript reports a study of how the ParB-parS partition complex assembles in bacteria. This complex is a major actor in genome (both chromosomes and plasmids) segregation. Homologous systems also position other complexes involved in chemotaxis, mobility etc., emerging as a global mechanism of cell patterning in bacteria, reinforcing the crucial importance of its understanding. The literature is currently rich, particularly with the recent discovery of a new and original switch in the ParB protein, using CTP binding/hydrolysis to close a DNA clamp at the parS sites, then open it after sliding for some distance. This has further strengthened the interest for ParB-parS assembly and how these new data can be incorporated into the previous models and fit the existing data is still a matter of debate. The current manuscript is a nice add on to this field since it provides a model integrating the known activities of ParB and fitting the data in the case of the ParB-parS complex of the model bacterium *C. crescentus*.

As a pure biologist, the technical aspect of the calculations is far beyond my knowledge. I have one major concern, which may result from the writing and organization of the manuscript or from a real scientific flaw. Along the text, the authors calculate parameters to feed their models from ChIP-seq data, assuming their hypotheses are true (i.e. the repartition of ParB dimers is due to sliding from parS sites, allowing calculation from the ChIP-seq data of e.g. the diffusion coefficient and the dissociation rate). By doing so, parameters are found that fit quite well the data, showing that the starting hypothesis is possible. This however does not rule out other hypotheses. For example, the ParB-ParB recruitment, when added into the model, makes it fitting less well. Have (or can) this mechanism been taken into account to recalculate the parameters ? Could a better fit be reached with ParB-ParB recruitment ? The authors should aim to make this clearer for the general reader or leave them with an impression of instruction in charge.

Indeed, other hypotheses are not ruled out. But to be clear, ParB sliding has been confirmed in vitro quite definitively and there is good in vivo evidence.

In the absence of bridging, ParB sliding necessarily and irrespective of parameters produces an exponentially decaying profile. This is also consistent with the experimental profile. With bridging, i.e. in the coupled simulations, we still observe this exponential decay.

However, when we add ParB-ParB recruitment, the decay changes and is no longer exponential - we refer to 'shoulders' at the extremes of the distribution. It can therefore not match the ChIP-Seq data. Adjusting the parameters did not change this. While we cannot check all parameter values of the coupled simulation, we do not believe that it can be made to fit due to the fundamentally different nature of the decay.

A model with only ParB-ParB recruitment and no sliding has also previously been considered and produces a linear decay (Broedersz 2014, Walter 2018) so again cannot fit the data.

The text has been updated to describe this more clearly.

Abstract : Is it useful to talk about speculation in the abstract ? There is no data supporting a role in SMC loading here, which is a pure hypothesis, thus welcome only in the discussion.

This is a fair criticism. We have removed it from the abstract.

Last paragraph of page 2 : the rationale about the degree of movement which may be different in the two regimes is difficult to follow. I have no solution to propose but the authors should attempt a clearer explanation.

We have amended this paragraph to try and make clearer the description of why these two regimes arise. Furthermore we have added a panel the SI figure to illustrate this description and show how long and short bridge lifetimes (in terms of the move rate of the polymer) can result in different polymer conformations.

Page 3 'short-lived ParB bridging leads to ...' : It is stated here that weak bridging leads to a regime with hairpins and helices. DNA supercoiling should have an important effect on the formation? dynamics? of these structures ? Can this be ignored ? It should at least be discussed better than just one sentence in the discussion section.

This should indeed have been discussed better. We expect that supercoiling and plectonemes in particular would only act to reinforce the formation of ParB hairpin structures. We have extended the discussion.

Explicitly simulating supercoiling would require a different type of simulation (Brownian dynamics) than we use here and so leave it for future work.

Fig1 : Is the red dot a parS site in panel A ? If so, why is there a single one here ? It would help to state early in the text and in the legend that two of the 5 parS are too close to be clearly distinguished in the figs.

We have amended Figure 1A to show 5 red dots representing the parS sites for consistency with the rest of the text. Earlier clarification has been added.

Fig2 : What are the dotted rectangles in panel A ? Does 'mons' state for 'monomers' in the Y-axis ? The shading is difficult to see in panel B.

We have removed the dotted rectangles from panel A. 'mons' has been changed to 'monomers' for clarity. We have changed the shading in panel B to show the SD rather than the SEM, this is broader and much easier to see.

Page 5 first paragraph : It seems to me that other authors have reported ChIP-seq patterns from a chromosome with multiple parS: *Vibrio*, others ? This may deserve to be cited here.

We have added a citation to the ChIP-seq data that has been measured in *Vibrio cholerae*.

Could the data be used to test the models presented here ?

In principle, yes. However, for *Caulobacter*, we have both measurements of the affinity of each parS site and FRAP measurements to fix k_{off} .

Third paragraph : FRAP data have been obtained with the F system. This should be cited and possible differences discussed.

We have now referenced the FRAP data for F-plasmid. The result is very similar to what we found (70s vs 64s) for *C. crescentus*.

Fourth paragraph, second sentence 'previous measurements' : please provide a reference for this.

Done.

Page 6 first paragraph of 'roadblocks' : the juggling between lifetime and % of occupancy is not obvious here for the non-specialist and only becomes clear when reading the figure and legend, please explain better.

We have added a further explanation of the difference between the lifetime and percentage occupancy of roadblocks to make this clearer.

Page 7 top : parameters (diffusion coefficient) need to be (and are) changed here to restore the observed distribution of ParB on the DNA. Why not later when the ParB-ParB recruitment is implemented ?

As described above, a change in parameters cannot rectify the issues posed by ParB-ParB recruitment.

'This suggests that sliding of ParB pushes the system towards the structure regime' : can the authors provide an (intuitive) explanation for this ?

We may have overstated things here. We now more precisely describe our result. In the coupled simulations, bridges are less efficient at condensing the partition complex, likely due to the roadblock effect of bridged ParB on sliding ParB making it less likely for the distal bridge that characterise that regime to form.

Last paragraph of the result section : There has been a problem with the text here (part is missing), please correct.

We apologise for this issue and it has been fixed.

REVIEWERS' COMMENTS

Reviewer #1 (Remarks to the Author):

I appreciate the authors' efforts; however, I'd have to say that they did not address the essential elements of my concerns over the physical basis and significance of the model. Maybe there is some misunderstanding. So let me explain my thoughts once more.

1. The authors first presented the two simplified models that interrogate the bridging and sliding processes respectively and separately. From these two simplified models, they fitted the relevant model parameters. They then combined the two models and argued that it is the coupling between ParB bridging and sliding that underlies the formation of PC complexes. The problem is: If the coupling between bridging and sliding is essential, then one cannot fit the model parameters for the sliding process to the experimental data, while ignoring the bridging process, and vice versa. This is because the experimental data itself embodies the coupling between bridging and sliding processes. Therefore, either the authors' point on the essential role of bridging-sliding coupling in PC formation holds up, or the parameter fitting of the two simplified models makes sense; but it cannot be true for both. This is a logic problem. The authors seem to follow a circular logic, which is problematic. To rescue and to avoid confusions, the two simplified models should be removed from the manuscript.

2. Related to 1): The authors assumed in the model that ParB spreading is independent of the 3D structure of DNA. I still think this assumption is problematic. For instance, the experiments in (ref#27, Jalal et al, eLife, 2020: <https://doi.org/10.7554/eLife.53515>) show that the DNA conformation does impact ParB-DNA binding, since a closed DNA substrate is required for an increased ParB association with DNA.

3. Another key model ingredient is that the authors assumed a roadblock effect from ParB complex along the DNA strand. The DNA strand is not a geometric line; instead, it has a physical dimension (~ 2 nm in diameter). In principle, two ParB complexes can occupy the same location along the length of the DNA (i.e., x-axis) but have different coordinates in the radial directions (i.e. y- and z-axis). As such, this assumption does not make physical sense. Now if one removes this assumption from the model, then the key conclusion of the manuscript will largely collapse. To rescue, the authors should provide the direct experimental evidence for the roadblock effect of DNA-bound ParB complexes. Otherwise, the model is on a shaky ground. Additionally, even if some location on DNA strand may not be bypassed via sliding, the ParB complexes can still bind to DNA from cytosol. Is there any experimental evidence showing that ParB complexes must slide along the DNA while they cannot bind to DNA from cytosol?

4. The relative timescale between DNA polymer dynamics and ParB bridging-sliding process holds the key for different DNA configurations, as the authors claimed. Under this context, in their rebuttal the authors stated “The intrinsic relaxation timescale of the DNA polymer in our model is determined by the overall move attempt rate parameter, p . Whilst this variable is not experimentally known the variation of it in relation to the ParB bridge lifetime is precisely what we probe in the first section of the paper where we show that different ParB bridge lifetimes (in units of $1/p$) result in distinctly different conformations...” Since the “ p ” is not experimentally known, the intrinsic timescale of DNA polymer dynamics is not constrained in the model. Moreover, there are no experiments in the paper that directly modulate the ParB-bridging timescale to test the model. From this sense, whatever the physical phenomena predicted by the model should be considered as hypothetical and may or may not be true. The authors should explicitly state the point in the paper so that the readers can better appreciate the work. Of course, this does not disapprove the model predictions. In the light of “theory can tell you what is physically possible, only experiments can tell you what really happens”, however, the lack of experimental support just dampens the scientific significance of the model as it currently stands. This adds to my initial concern about the significance of this work, considering that much has been done in the previous works by others.

5. To faithfully simulate the dynamical coupling between ParB bridging and sliding processes, in revision the authors seemed to carry out the true kinetic implementation of the model, instead of the coupled simulation as they termed it. As the authors stated in the rebuttal “...The only difference between the coupled simulations and a true kinetic implementation is that new bridging events occur based on the ParB locations at the end of previous iteration of the sliding simulation. This is a reasonable simplification as long as the sliding ParBs equilibrate faster the bridging timescale, which we expect to be the case given the ~ 1 min turnover. In any case, in response to Reviewer 2, we have now provided a true kinetic coupling of bridging and sliding. Our results are unchanged.” If the sliding timescale is much faster than the bridging timescale, then how could they couple? I apologize that I may miss something, but I am confused here: I thought the key conclusion is that the bridge-sliding coupling is essential for PC formation. And yet, in the revision the “true kinetic coupling of bridging and sliding” simulation – that shows the same/similar result as the coupled simulation in the original manuscript – only holds up when the sliding is much faster than bridging. The latter condition is for the decoupling between the sliding and bridging processes, isn't it?

Reviewer #2 (Remarks to the Author):

The authors have addressed my criticisms and comments, and I believe that the manuscript is now ready for publication. In particular, I think that this is a key work in the development of models aimed at understanding the phenomenology of Par(A)BS systems.

Reviewer #3 (Remarks to the Author):

The manuscript has been significantly improved, particularly concerning clarity for the non-specialist readers. Of evidence, other parameters such as DNA super-coiling will have to be considered, although I agree this is beyond the scope of the present study. The authors have answered my concerns and comments and, from my point of view and expertise, the manuscript can now be published. The advice of the other two reviewers, who are specialists of modelling, should however be considered as a priority.

REVIEWER 1's COMMENTS:

I appreciate the authors' efforts; however, I'd have to say that they did not address the essential elements of my concerns over the physical basis and significance of the model. Maybe there is some misunderstanding. So let me explain my thoughts once more.

1. The authors first presented the two simplified models that interrogate the bridging and sliding processes respectively and separately. From these two simplified models, they fitted the relevant model parameters. They then combined the two models and argued that it is the coupling between ParB bridging and sliding that underlies the formation of PC complexes. The problem is: If the coupling between bridging and sliding is essential, then one cannot fit the model parameters for the sliding process to the experimental data, while ignoring the bridging process, and vice versa. This is because the experimental data itself embodies the coupling between bridging and sliding processes. Therefore, either the authors' point on the essential role of bridging-sliding coupling in PC formation holds up, or the parameter fitting of the two simplified models makes sense; but it cannot be true for both. This is a logic problem. The authors seem to follow a circular logic, which is problematic. To rescue and to avoid confusions, the two simplified models should be removed from the manuscript.

We never argue that "it is the coupling between ParB bridging and sliding that underlies the formation of PC complexes". Note the word 'can' in the title of our paper. Our model does not seek to propose new interactions that have no experimental support. Both ParB bridging and its ability to slide along the DNA have been determined in *in vitro* studies and *in vivo* work has indicated the importance of these processes in partition complex formation or dynamics (Graham et al 2014 Genes&Dev, Osorio-Valeriano et al 2021, Mol.Cel). Our work explores the effect of these two processes, whether they are compatible, and can reproduce the observed binding profile and partition complex compaction. The question of compatibility is important as bridged ParB would act as roadblocks for sliding ParB and this is the point of our coupled simulations. We are clear about this aim. Note the first sentence of the coupling section: "We next investigate whether ParB bridging is compatible with the ParB binding profile ...".

Regarding parameter fitting, we do not have data to constrain all the parameters of the full model but we never claim to have a completely constrained model (again note the word 'can' in the title). It should be noted that for the coupled model we arbitrarily choose two values of p to represent the globule-like and structured regime (based on the sweep of our simple bridging model) and then refit for the ParB bridging rate and diffusion coefficient. The ParB dissociation rate is estimated experimentally (FRAP). So only the ParB binding rate is carried over from the fitting to the sliding-only model. This could have been better described in the text and we will address this upon revision. Despite the fact we are unable to entirely constrain the full model we nevertheless present it in the text to illustrate that the processes of sliding and bridging can coexist and reproduce the observed ParB binding profile and distribution.

2. Related to 1): The authors assumed in the model that ParB spreading is independent of the 3D structure of DNA. I still think this assumption is problematic. For instance, the

experiments in (ref#27, Jalal et al, eLife, 2020: <https://doi.org/10.7554/eLife.53515>) show that the DNA conformation does impact ParB-DNA binding, since a closed DNA substrate is required for an increased ParB association with DNA.

Bacterial chromosomes and plasmids are typically circular (at least all for all the ParABS systems that we know of). ParB sliding off the end of a DNA strand as shown in Jalal et al is not likely to be relevant in vivo. Comparing open vs closed DNA strands was used to test ParB sliding. While our polymer simulations use a linear polymer (chosen to obtain the correct polymeric behaviour of a 10kb section of a 4Mb circular chromosome), it is long enough that ParB dissociates before it reaches the end. We know of no study showing an effect of the actual DNA conformation, as opposed to DNA topology (circular/open), on ParB-DNA binding.

3. Another key model ingredient is that the authors assumed a roadblock effect from ParB complex along the DNA strand. The DNA strand is not a geometric line; instead, it has a physical dimension (~ 2 nm in diameter). In principle, two ParB complexes can occupy the same location along the length of the DNA (i.e., x-axis) but have different coordinates in the radial directions (i.e. y- and z-axis). As such, this assumption does not make physical sense. Now if one removes this assumption from the model, then the key conclusion of the manuscript will largely collapse. To rescue, the authors should provide the direct experimental evidence for the roadblock effect of DNA-bound ParB complexes. Otherwise, the model is on a shaky ground. Additionally, even if some location on DNA strand may not be bypassed via sliding, the ParB complexes can still bind to DNA from cytosol. Is there any experimental evidence showing that ParB complexes must slide along the DNA while they cannot bind to DNA from cytosol?

As we stated in the text and in our response, ParB can entrap the DNA: ParB dimers form a closed protein clamp that completely circumvents the DNA strand (references in the text). Therefore two ParB dimers can not occupy the same location along the length of the DNA.

The roadblock effect of DNA binding proteins (that do not circumvent the DNA) on ParB sliding has been clearly shown (this was one of the main findings of Jalal et al cited above). The same study and others (Osorio-Valeriano et al 2019 Cell,) saw no significant ParB loading onto the DNA in the absence of the parS site i.e. binding 'from the cytosol' is likely not relevant (however there is some evidence for ParB-ParB recruitment independent of parS that we discuss).

4. The relative timescale between DNA polymer dynamics and ParB bridging-sliding process holds the key for different DNA configurations, as the authors claimed. Under this context, in their rebuttal the authors stated "The intrinsic relaxation timescale of the DNA polymer in our model is determined by the overall move attempt rate parameter, p . Whilst this variable is not experimentally known the variation of it in relation to the ParB bridge lifetime is precisely what we probe in the first section of the paper where we show that different ParB bridge lifetimes (in units of $1/p$) result in distinctly different conformations..." Since the " p " is not experimentally known, the intrinsic timescale of DNA polymer dynamics is not constrained in the model. Moreover, there are no experiments in the paper that directly modulate the ParB-bridging timescale to test the model. From this sense, whatever the

physical phenomena predicted by the model should be considered as hypothetical and may or may not be true. The authors should explicitly state the point in the paper so that the readers can better appreciate the work. Of course, this does not disapprove the model predictions. In the light of “theory can tell you what is physically possible, only experiments can tell you what really happens”, however, the lack of experimental support just dampens the scientific significance of the model as it currently stands. This adds to my initial concern about the significance of this work, considering that much has been done in the previous works by others.

As mentioned previously we are clear throughout the paper that we simply explore the effect of ParB sliding and bridging and their ability to reproduce the experimental observations. Regarding the bridge lifetime, since we do not know p , or indeed the bridge lifetime itself, the ratio of the bridge lifetime and the polymeric timescale $1/p$ is an unknown quantity. That is why we explore both of the regimes identified and perform coupled simulations at a nominal location within each regime. We fully acknowledge that the hairpins of the structured regime may not in fact be biologically relevant, however in the absence of decisive experimental measurements, it is still informative to explore what conformations might be generated by ParB sliding and bridging. Furthermore, the question of whether ParB sliding and bridging are compatible and can reproduce experimental observations is important and has not previously been studied.

5. To faithfully simulate the dynamical coupling between ParB bridging and sliding processes, in revision the authors seemed to carry out the true kinetic implementation of the model, instead of the coupled simulation as they termed it. As the authors stated in the rebuttal “...The only difference between the coupled simulations and a true kinetic implementation is that new bridging events occur based on the ParB locations at the end of previous iteration of the sliding simulation. This is a reasonable simplification as long as the sliding ParBs equilibrate faster the bridging timescale, which we expect to be the case given the ~1min turnover. In any case, in response to Reviewer 2, we have now provided a true kinetic coupling of bridging and sliding. Our results are unchanged.” If the sliding timescale is much faster than the bridging timescale, then how could they couple? I apologize that I may miss something, but I am confused here: I thought the key conclusion is that the bridge-sliding coupling is essential for PC formation. And yet, in the revision the “true kinetic coupling of bridging and sliding” simulation – that shows the same/similar result as the coupled simulation in the original manuscript – only holds up when the sliding is much faster than bridging. The latter condition is for the decoupling between the sliding and bridging processes, isn't it?

There seems to be some confusion here. The quotation provided refers to the simplification we used in the first version of the coupled model (we use ‘coupled’ simply in the sense of combined i.e. simulations with both sliding and bridging). In the revised model, we have performed a proper kinetic implementation of the two processes - no assumptions are made in the implementation.

Again, note that we do not claim that “the bridge-sliding coupling is essential for PC formation”. Rather we show that sliding and bridging are compatible and together can, in principle, reproduce both partition complex compaction and the ParB binding profile. Of course, there are constraints. If bridges are too long-lived compared to the timescale of

sliding then sliding is inhibited and the profile is not reproduced. But our point is that the experimental observation can, in principle, be reproduced by sliding and bridging. Note again the word 'can' here and in the title. Of interest is also our observation of the induced hairpin structures and our speculation of their role in SMC loading.